# Monkeys exhibit human-like gaze biases in economic decisions

**Shira M Lupkin[1,2], Vincent B McGinty[1]\***

[1]Center for Molecular and Behavioral Neuroscience, Rutgers University, Newark, United States; [2]Behavioral and Neural Sciences Graduate Program, Rutgers University, Newark, United States

**Abstract** In economic decision-making individuals choose between items based on their perceived value. For both humans and nonhuman primates, these decisions are often carried out while shifting gaze between the available options. Recent studies in humans suggest that these shifts in gaze actively influence choice, manifesting as a bias in favor of the items that are viewed first, viewed last, or viewed for the overall longest duration in a given trial. This suggests a mechanism that links gaze behavior to the neural computations underlying value-based choices. In order to identify this mechanism, it is first necessary to develop and validate a suitable animal model of this behavior. To this end, we have created a novel value-based choice task for macaque monkeys that captures the essential features of the human paradigms in which gaze biases have been observed. Using this task, we identified gaze biases in the monkeys that were both qualitatively and quantitatively similar to those in humans. In addition, the monkeys' gaze biases were well-explained using a sequential sampling model framework previously used to describe gaze biases in humans—the first time this framework has been used to assess value-based decision mechanisms in nonhuman primates. Together, these findings suggest a common mechanism that can explain gaze-related choice biases across species, and open the way for mechanistic studies to identify the neural origins of this behavior.

**\*For correspondence:**
VINCE.MCGINTY@RUTGERS.
EDU

**Competing interest:** The authors declare that no competing interests exist.

## Editor's evaluation

This study analyzed viewing behavior in monkeys during value-based decision-making to determine whether relationships between gaze patterns and choices previously described in humans are also present in monkeys. The study used a clever task design and sophisticated modeling approaches to reveal robust evidence for similarities to extant human data. This is important to the field because it suggests common neural mechanisms linking viewing behavior and decision-making, which can now be further explored across species.

## Introduction

Economic decisions are ubiquitous in the natural world, and in natural settings humans and other primates often evaluate the available options by moving their eyes, rapidly shifting the focus of their gaze between the options as they deliberate. Over the past decade, studies in humans have documented an association between gaze and value-based choices. In particular, when offered a choice between multiple items, people are more likely to select the first item that they view, the item they look at just prior to indicating their decision, and the item that they spend the longest time viewing over the course of their deliberation. Further, choice biases can be induced through manipulations of gaze (*Gwinn et al., 2019*; *Liu et al., 2020*; *Liu et al., 2021*; *Pärnamets et al., 2015*; *Sui et al., 2020*; *Tavares et al., 2017*). Together, these studies suggest that gaze plays an active role in the decision

**eLife digest** When we choose between two items, we might expect to spend more time looking at the one we have a pre-existing preference for. For example, at the grocery store, you might assume that someone who likes grapes better than bananas would spend a longer time looking at the grapes. Surprisingly, a series of studies on human decision-making have shown that the opposite relationship is also true: the more time we spend looking at an item, the more likely we are to pick it. This 'gaze bias' occurs in many real-life and laboratory decision settings, and it is especially evident for choices between two equally preferred options. However, examining the brain circuits that underpin this behavior has so far been difficult due to a lack of animal models in which to study them.

In response, Lupkin and McGinty proposed that rhesus macaques may be the ideal species in which to study gaze biases, as these animals likely rely on the same brain regions as humans when gazing and making decisions. To test this hypothesis, a computer-based decision game similar to the ones used for humans was designed for the monkeys. It involved the animals having to choose between two icons that were associated with different amounts of a juice reward. Analysing how long the macaques had spent looking at each icon before making their choice revealed that they indeed tended to select the icon they had looked at for longer – including when the two icons indicated equal rewards. Other types of gaze biases present in humans were also detected, such as choosing the icon that was viewed first or last in a trial.

Additional analyses using computer simulations confirmed that the gaze biases of humans and monkeys were comparable and, critically, that they could be explained by similar underlying brain processes.

These strong similarities suggest that rhesus macaques could be used to study the neural basis for decision-making in both humans and nonhuman primates, potentially making it easier to examine the harmful changes in decision-making that occur in conditions like substance abuse or depression.

process, such that simply viewing a given option makes us more likely to choose it (see *Krajbich, 2019*, for a review).

However, despite increased interest in the role of gaze in decision-making, knowledge of the underlying neural mechanisms is still limited (c.f., *Krajbich et al., 2021*; *Lim et al., 2011*). One critical barrier to identifying these mechanisms is the lack of a suitable animal model of this behavior, and consequently a lack of studies addressing neural mechanisms at the cellular level. To this end, the aim of the present study was twofold: first, to develop a novel behavioral paradigm for macaque monkeys that captures the essential features of tasks previously used in humans; and second, to determine whether monkeys exhibit gaze-based choice biases that are similar to those of humans.

While the neural mechanisms linking gaze and economic choice are not fully known, the underlying computations have been explored using sequential sampling models, a highly successful class of models used to explain simple decision behaviors (*Ratcliff and McKoon, 2008*). In these models, a stream of noisy evidence for each option is accumulated until one item accrues enough evidence to reach a pre-defined threshold, and a decision is rendered in favor of that item. For value-based tasks, the evidence accumulation rate can be formulated as a function of the relative value of the possible outcomes—the larger the difference in relative value, the faster that evidence accumulates for the higher valued item. Over the last decade, the basic sequential sampling model framework has been extended to account for the influence of gaze behavior (i.e. overt attention) on the decision process. These models, collectively referred to as attentional sequential sampling models (aSSMs), assume that evidence for a given item accumulates at a faster rate while that item is being viewed, and accumulates slower when it is not being viewed (*Krajbich et al., 2010*; *Krajbich and Rangel, 2011*; *Smith and Krajbich, 2018*; *Tavares et al., 2017*; *Thomas et al., 2019*). Overall, these models have been successful in accounting for human gaze biases across several decision-making domains (*Krajbich, 2019*; *Manohar and Husain, 2013*; *Smith and Krajbich, 2018*; *Thomas et al., 2019*; *Westbrook et al., 2020*), supporting the idea that gaze modulates the decision process by biasing an evidence accumulation process.

To identify candidate neural substrates, intracranial recordings in nonhuman primates (NHPs) are an ideal approach, as they combine an animal model with human-like oculomotor behavior with the high

temporal resolution necessary to accommodate the rapid pace of natural eye movements (*Kravitz et al., 2013*; *Ongür and Price, 2000*; *Wise, 2008*). Moreover, NHPs are commonly used as animal models in neuroeconomics. However, existing paradigms contain elements that make them unsuitable for testing the relationship between gaze and choice behavior. More specifically, many NHP studies restrict eye movements and/or use eye movements as a means of reporting choice, rather than for inspecting the choice options, (*Cavanagh et al., 2019*; *Hunt et al., 2018*; *Rich and Wallis, 2016*). These common restrictions can interfere with natural gaze patterns and therefore limit the ability to identify the relationship between gaze behavior and choice.

To address the need for an animal model of gaze biases, we have developed a novel behavioral paradigm in which monkeys are able to freely deploy their gaze to the targets on the screen and to indicate their choice by a manual response. Using this paradigm, we replicated the three core behavioral signatures of gaze-related choice biases: We found that the monkeys were more likely to choose the first item that they viewed, the last item that they viewed before deciding, and the items that received more overall gaze time. In addition, we use a computational model previously used to assess human gaze biases (*Thomas et al., 2019*) to determine whether similar computational mechanisms can explain choices in NHPs.

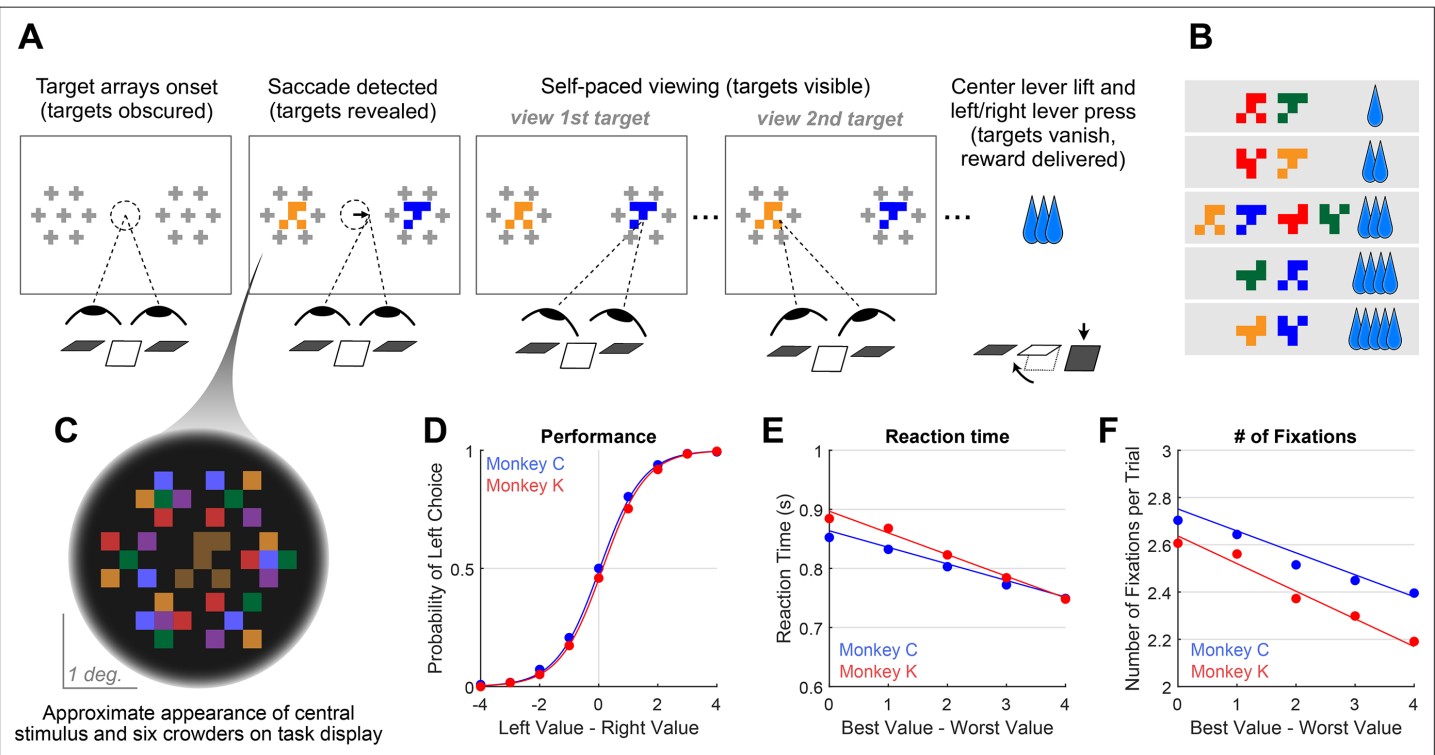

**Figure 1.** Decision-making task and performance in two monkeys. (**A**) Abbreviated task sequence; see *Figure 1—figure supplement 1* for full task sequence. The interval between target onset and center lever lift defines the decision reaction time (RT). The yellow and blue glyphs are choice targets, and the gray '+' shapes indicate the location of visual crowders designed to obscure the targets until they are viewed (fixated) directly. For clarity, in this panel crowders are shown in gray and at a reduced scale; on the actual task display, crowders were multicolored and the same size as the targets, as in panel C. (**B**) An example set of 12 choice targets, organized into 5 groups corresponding to the 5 levels of juice reward. (**C**) Close-up view of a single target array, consisting of a central yellow choice target surrounded by six non-task-relevant visual crowders. Two such arrays appear on the display (panel A), each located 7.5° from the display center. (**D–F**). Task performance. (**D**) Fraction of left choices as a function of the left minus right target value (in units of juice drops). (**E**) Reaction Time (RT) decreases as a function of difficulty, defined as the absolute difference between target values; (**F**) Number of fixations per trial decreases as a function of difficulty. Filled circles show the mean values and smooth lines show the logistic (**D**) or linear (**E–F**) regression fits derived from mixed effects models (*Supplementary file 1*, rows 1–3, respectively). Error bars are too small to be plotted. The data in blue are from Monkey C (N = 15,613), and in red, from Monkey K (N = 14,433).

The online version of this article includes the following figure supplement(s) for figure 1:

**Figure supplement 1.** Full task sequence.

## Results

### Basic psychometrics

We trained two monkeys to perform a value-based decision task in which they used eye movements to view the choice options and used a manual response lever to indicate their decision (*Figure 1A*). To begin a trial, the monkeys were required to fixate on the center of the task display while holding down the center of three response levers. After the required eye fixation and lever hold period was satisfied, two targets were presented on the left and right of the task display, each associated with a reward ranging from 1 to 5 drops of juice. Following the initial fixation period, gaze was unrestricted, allowing the monkeys to look at each target as many times and for as long as they liked before indicating a choice. To encourage the monkeys to look directly at the targets, and to limit their ability to perceive the targets using peripheral vision, the targets were initially masked until the first saccade away from the central fixation point was detected (see *Methods*). To indicate their choice, the monkeys first lifted their hand off the centrally-located response lever, and then pressed the right or left lever to indicate which of the two targets they chose. The reaction time (RT) in each trial was defined as the period between the onset of the targets and the lift of the center lever. We collected data from a total of 54 sessions: 29 from Monkey C (N = 15,613) and 25 for Monkey K (N = 14,433).

Overall choice performance is illustrated in *Figure 1D*. Choices were nearly optimal for the easiest trials (value difference of +/-3 or 4 drops of juice, >98% of choices in favor of higher value target) and occasionally suboptimal for more difficult trials (value difference of +/-1 or 2 drops of juice, 86% higher-value choices for both Monkey C and Monkey K). When the targets were of equal value, the probability of the monkeys choosing the left or right target was near chance (50% left choices for Monkey C, 46% for Monkey K). This pattern of choices indicates that the monkeys were behaving according to the incentive structure of the task, with a very low rate of guesses or lapses (*Fetsch, 2016*). Further, choice behavior was well-described by a logistic function parameterized by the difference between the values of the left and right items (fit lines in *Figure 1D* are from a logit mixed effects regression: $\beta_{value-difference}$ = 1.37, CI = [1.34, 1.40]; $F_{(1,30044)}$=8411.60, p<1e-10). Additionally, we found that both the average RT and the average number of fixations per trial were modulated by choice difficulty: the monkeys responded slower and made more fixations when the two targets were near in value, compared to when the value difference between the targets was large (RT: *Figure 1E*, linear mixed effects regression: $\beta_{difficulty}$ = -0.032, CI: [-0.039,–0.026]; $F_{(1, 30044)}$=106.09, p<1e-10. Number of Fixations: *Figure 1F*, Linear mixed effects regression: $\beta_{difficulty}$ = -0.10, CI = [-0.12,–0.09]; $F_{(1, 30044)}$=126.7, p<1e-10). Together these results are consistent with behavioral patterns seen in both humans and NHPs performing 2-alternative forced choice tasks (e.g. *Gold and Shadlen, 2007*; *Thomas et al., 2019*).

### Effects of difficulty and chosen value on reaction times and the number of fixations

In the preceding section, we show that both RT and the number of fixations increase as a function of trial difficulty. However, in high-performing subjects trial difficulty is correlated with chosen value, because the trials with large value differences (low difficulty) are also trials where higher-value offers are chosen. Therefore, the RT and fixation effects originally attributed to difficulty may be explained in part by chosen value. To determine the unique contributions of chosen value and difficulty to RT and the total number of fixations, we used the approach of *Balewski et al., 2022*. Briefly, we modeled each behavioral metric as a function of one predictor, and then regressed the other predictor against the residuals of the first model. For example, to measure the unique contribution of chosen value to RT, we first ran a linear mixed effects model explaining RT as a function of difficulty. We then took the residuals from this model, which represent the RT after accounting for difficulty, and ran a second mixed effects model explaining these residuals as a function of chosen value.

Consistent with the findings of Balewski and colleagues, RT was modulated more by chosen value than by difficulty ($\beta_{chosen-value}$ = -0.026, CI = [-0.034,–0.019]; $F_{(1,30044)}$=46.35, p<1e-10; CPD = 8%; $\beta_{difficulty}$ = -0.011, CI = [-0.012,–0.010]; $F_{(1,30044)}$=318.44, p<1e-10; CPD = 1.4% *Figure 2A/C*, *Supplementary file 1* rows 4–5). Interestingly, we found the opposite pattern for the total number of fixations per trial: this metric was strongly modulated by difficulty ($\beta_{difficulty}$ = -0.065, CI = [-0.086,–0.043]; $F_{(1,30044)}$=34.46, p=4.41e-9; CPD = 1.8%; *Figure 2C*), and showed no effect of chosen value ($\beta_{chosen-value}$ = -0.028, CI = [–0.095, 0.39]; $F_{(1,30044)}$=0.66, p=0.42; CPD = 0.69%; *Figure 2D*, *Supplementary file 1* rows 4–5). In

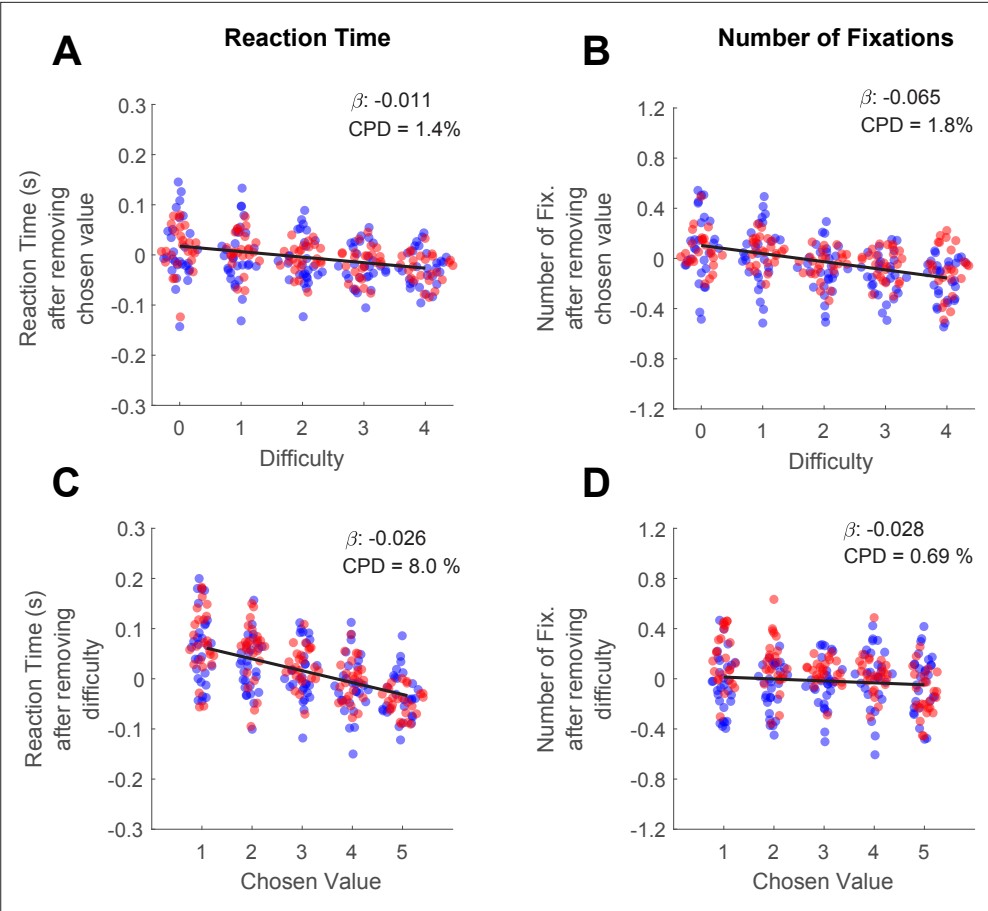

**Figure 2.** Differential contributions of difficulty and chosen value to decision reaction times and the number of fixations per trial. (**A**) The effect of trial difficulty on reaction time after accounting for the effect of chosen value. (**B**) The effect of trial difficulty on number of fixations after accounting for chosen value. (**C**) The effect of chosen value on reaction time after accounting for difficulty. (**D**) The effect of chosen value on number of fixations after accounting for difficulty. Y-axes give the dependent measures (RT or number of fixations) after being residualized by either difficulty or chosen value, as indicated (see Methods). Dots indicate the mean of each dependent measure for each session. Black lines show linear model fits to the plotted dots. The betas and coefficients of partial determination (CPD) of these models are in the top right corner. Blue dots show the session means from Monkey C (N=25 sessions, 15,613 trials), red dots are from Monkey K (N=29 Sessions,14,433 trials).

sum RTs appear to reflect the size of the upcoming reward (faster responses for higher rewards), whereas the number of fixations reflect choice difficulty, with more fixations associated with more difficult trials.

## Gaze behavior and basic fixation properties

The following sections characterize the number and durations of the monkeys' fixations onto the choice targets throughout each trial. A fixation was defined as a period of stable gaze upon one of the two targets. Consistent with fixation analyses in human studies, consecutive fixations upon the same target were merged into a single fixation epoch, and fixations onto non-target locations were not included in the analysis (See *Behavior Analysis* for details). Following conventions from human studies, final fixation durations are defined as the interval between fixation onset and the center lever lift; in other words, final fixation durations are delimited by the reaction time in each trial (e.g. *Krajbich et al., 2010*). For non-final fixations, the duration is defined in the conventional way, as the interval between fixation onset and the initiation of the next saccade.

## Number of fixations per trial

In our task, information about the choice targets was obscured unless the monkey was directly fixating on them. Therefore, the optimal strategy involves a minimum of two fixations, one to each target.

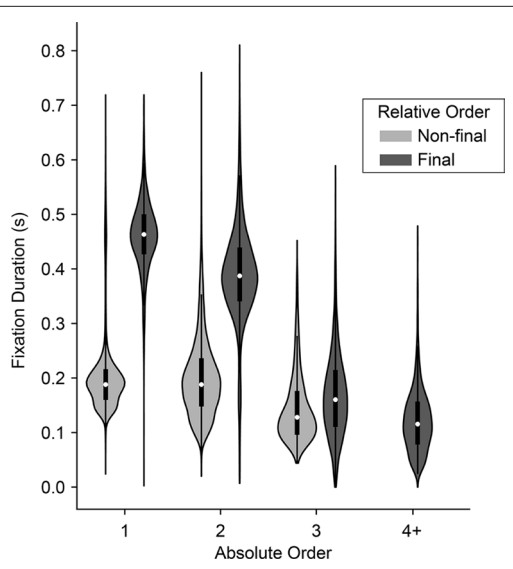

**Figure 3.** Distribution of fixation durations across absolute and relative position in the trial. Violins show distribution of fixation durations across absolute position in the trial (x-axis) and split according to whether they were final (dark gray) or non-final (light gray) fixations in the trial. 1st fixations: N=30,046 non-final and 1,652 final; 2nd fixations: N=15,753 non-final and 12,641 final; 3rd fixations: N=1,749 non-final and 14,004 final; 4th fixations or greater: N=80 non-final (not shown) and 1,749 final.

Consistent with this, the monkeys almost always looked at both targets on each trial (97.2% of trials for Monkey C, 91.6% for Monkey K). The fraction of trials with 1, 2, 3, or ≥4 total fixations before the RT for Monkey C was 2.8%, 41.2%, 49.2%, and 6.8%; and for Monkey K was 8.4%, 43.0%, 43.8%, 4.8%. Thus, in the vast majority of trials the monkeys either looked at each target exactly once (2 total fixations) or looked at one target once and the other twice (3 total fixations). The average number of fixations per trial for Monkey K was 2.45 SEM 0.02 and for Monkey C was 2.57 SEM 0.04. This average is smaller than in human binary choice tasks, in which the average number of fixations per trial is usually 3 or more. As we discuss below, the difference between human and monkey total fixation numbers has implications for comparing fixation durations between species.

## Fixation durations depend on relative and absolute order in the trial

Fixations can be defined by both their absolute and relative order in a trial. Absolute order refers to the serial position in the trial: first, second, third, etc. Relative order refers to when the fixation occurs with regards to the trial start or end: initial, middle, final. Note that these definitions are not independent or mutually exclusive. Instead, they depend on the total number of fixations in the trial. For example, in a trial with a total of two fixations, the second fixation is also the 'final' fixation (there are no middle fixations). However, in a trial with three total fixations, the second fixation is considered a 'middle' fixation. Because the monkeys usually make either two or three total fixations in a trial, relative and absolute order are conflated. (In contrast, humans typically make three or more fixations per trial, so that second fixations are almost always 'middle' fixations.) Consequently, it is necessary to independently analyze the effects of both absolute and relative fixation order.

To assess the effects of absolute order (first, second, third, etc.) independent of relative order (final/non-final), we first fit a linear regression explaining fixation duration as a function of relative order. We then regressed absolute order against the residuals from the first model. Consistent with human fixation patterns, we found that fixation durations decreased as a function of absolute order, after controlling for relative order ($\beta_{relativeOrder}$ = -0.049, CI = [-0.065,–0.033]; $F(1, 7762)$=36.06, $p$<1.92e-9; *Figure 3*). The decrease is especially notable at the third fixation, which in a binary task is the first 're-fixation' onto a previously sampled target, and is consistent with human fixation patterns (*Manohar and Husain, 2013*). Next, to assess the effects of relative order independent of absolute order, we fit a linear mixed-effects model comparing fixation durations across absolute positions, treating absolute position as a categorical variable. We then fit a second model explaining the residuals as a function of whether or not the fixation was a final fixation (i.e., relative order). After accounting for serial position, we found that non-final fixations were shorter than final fixations ($\beta_{relativeOrder}$ = -0.047, CI = [-0.057,–0.037]; $F(1, 7762)$=86.34, $p$<1e-10; *Figure 3*). In sum, we identified two independent order-based effects on duration: a decrease in duration with successive fixations in a trial, as well as a tendency to fixate longer on final fixations.

The monkeys' prolonged final fixations seem to contradict prior human studies, which typically report shorter final fixations. However, this discrepancy may be explained by the relatively larger number of fixations that humans make (mean ~3.7, computed from freely available data from the

study described in *Krajbich et al., 2010*), compared to the monkeys in the present study (mean ~2.5). Consequently, humans' final fixations tend to have higher absolute order and are thus nearly always re-fixations onto a target—both factors that are associated with shorter fixation durations, as noted above. In contrast, the monkeys' final fixations are overwhelmingly either the second or third in the trial. Thus, not only do their final fixations have a lower absolute order, but they are also comprised of a larger fraction of longer initial fixations onto a target (i.e. when the final fixation is the 2nd fixation).

## Fixation durations depend on target value

The average fixation duration was dependent on the value of the target being fixated, and upon the order in the trial in which the fixation occurred. For the first fixation in each trial, duration increased as a function of fixated target value ($\beta_{fixValue}$ = 0.003, CI = [0.0011,0.0040]; F(1, 28392)=11.37, p=0.0007, *Supplementary file 1*, row 8). Intuitively, the effect size can be interpreted as the influence of a one-drop change in juice volume; for example, for initial fixations the regression estimate of 0.003 corresponds to a 3ms increase in fixation duration per drop of juice. For middle fixations, defined as any fixation that was neither first nor final, the average duration decreased as a function of the value of the fixated target ($\beta_{fixValue}$ = -0.0076, CI = [-0.0086,–0.0067]; F(1, 17580)=231.37, p<1e-10; *Supplementary file 1*, row 9). At the same time, middle fixation durations were also influenced by the value of the target not currently being viewed ($\beta_{nofixValue}$ = -0.035, CI=[-0.040,–0.030]; F(1, 17580)=197.78, p<1e-10; *Supplementary file 1*, row 10); because most middle fixations were the second fixations in the trial, this means that the value of the first-viewed item 'carried over' into the second fixation to influence its duration. Combining the influences of fixated and non-fixated values into a single variable, middle fixation durations increased as a function of relative value, defined here as the value of the fixated target minus the value of the non-fixated target ($\beta_{relativeFixVal}$ = 0.016, CI = [0.012, 0.021]; F(1, 17580)=48.55, p<1e-10; *Supplementary file 1*, row 11). Finally, we find that durations decreased as function of the absolute difference in target values ($\beta_{difficulty}$ = -0.019, CI = [-0.025,–0.0154]; F(1, 17580)=47.17, p<1e-10; *Supplementary file 1*, row 12). Together, these results support the notion that fixation behavior and valuation are not independent processes, and that the effects of value on viewing durations must be accounted for when computing and interpreting gaze biases (see *Cumulative Gaze-Time Bias*, below).

## Gaze-related choice biases

Humans show three core choice biases related to gaze behavior. First, they tend to choose the item that they viewed first (initial fixation bias). Second, they tend to choose the item they spend more time looking at over the course of the trial (cumulative gaze-time bias). Third, they tend to choose in favor of the target they are viewing at the moment they indicate their choice. All three of these biases are evident in our monkey subjects, as detailed below. However, before these biases can be quantified, we must first address the question of whether monkeys show evidence for a two-stage form of gaze bias recently identified in human behavior.

## Evidence for a two-stage model of gaze biases

To explain how these gaze biases come about, prior human studies have used modified sequential sampling models (aSSMs) that provide an increase in evidence accumulation for a given item while it is being viewed (see *Krajbich, 2019*, for review). This bias generally takes one of two forms: Some studies suggest that the gaze amplifies the value of attended targets so that the increase in evidence accumulation is proportional to the attended value (i.e. is multiplicative). Others suggest that gaze acts by providing a fixed increase in evidence accumulation to attended targets, independent of target values (i.e. is additive). Studies directly comparing these two mechanisms are not fully conclusive (*Smith and Krajbich, 2019*; *Thomas et al., 2019*), perpetuating debates as to which is 'correct'. A third, hybrid modeling framework offers a compromise (*Manohar and Husain, 2013*; *Westbrook et al., 2020*). Under this framework, a trial can be divided into two stages. The first stage is the decision stage, where evidence accrues for the decision offers, and gaze increases evidence accumulation for attended items in multiplicative fashion. The second stage begins once an internal decision boundary is reached (a choice is made) and continues until the choice is physically reported. During this interval the gaze becomes drawn to the to-be-chosen item, which is captured with an additive

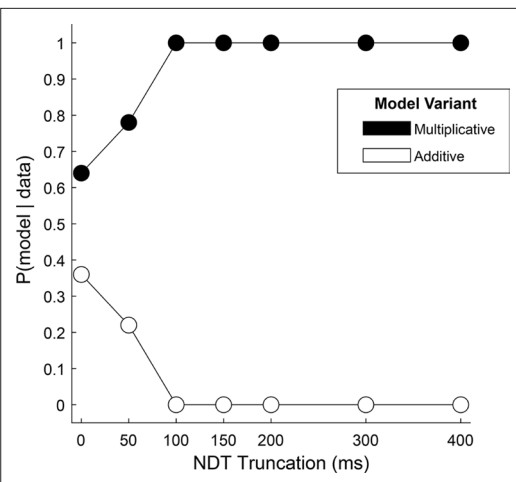

**Figure 4.** Relative fit of multiplicative and additive models across NDT truncation values. Using an aSSM modeling framework, we computed the posterior probability (y-axis) of a model with either a multiplicative gaze bias parameter (black circles) or an additive parameter (white circles) being more likely to have generated the data. Probabilities (x-axis) were computed using data in which the final portion of the trial was truncated by 0ms (N = 30,046 trials), 50ms (N = 30,045 trials), 100ms (N = 30,045 trials), 150ms (N = 30,044 trials), 200ms (N = 30,044 trials), 300ms (N = 30,011 trials), and 400ms (N = 29,771 trials). See *Estimating and truncating the terminal non-decision time* for details.

gaze bias. In other words, in the first stage, gaze influences choice, whereas in the second stage, the latent choice influences gaze.

Two patterns in the monkeys' fixation data are consistent with this two-stage model. First, as shown in *Figure 3*, final fixation durations are longer than non-final fixations even after accounting for absolute order in the trial, suggesting some process that occurs at the end of each trial that prolongs fixation durations. Second, the monkeys' final fixations were on the to-be-chosen target on nearly every trial (~97% of trials), consistent with gaze being drawn to the to-be-chosen items.

To more directly test the hybrid, two-stage, model in our data, we reasoned that the relative fit of a model with an additive gaze bias compared to one with a multiplicative gaze bias would indicate whether choice reflects one mechanism alone, or a mixture of both. Using the Gaze-Weighted Linear Accumulator Model (GLAM, *Thomas et al., 2019*), we found that while the multiplicative model provided a slightly better fit than the additive model, the difference was not significant (*Figure 4*, 0 on the x-axis, *Supplementary file 3*). This suggests that our data likely reflect a mixture of additive and multiplicative gaze biases, consistent with the hybrid model.

Next, we tested the assumption that the influence of gaze switches between the two biasing mechanisms near the end of the trial, at the moment a decision threshold is met. We reasoned that if this were the case, that the relative fit of one model over the other would increase if we removed increasing amounts of data from the end of the trial. Alternatively, if the multiplicative and additive influences were mixed throughout the trial, this truncation would have little effect on the relative fit of the two model variants. We found that the difference in fit between the two models became significant for truncation values of 100ms and higher, with the multiplicative model providing the better fit (*Supplementary file 3*). Further, at 100ms the probability that data were generated by a multiplicative, and not an additive, model reaches 100% (*Figure 4*). Together, this supports the two-stage model put forth by Westbrook, where gaze acts multiplicatively during the first stage before transitioning to additive approximately 100ms before the RT.

## Estimating and truncating the terminal non-decision time

Under the two-stage framework, the transition point between the multiplicative and additive stages of the decision process occurs when accumulated evidence reaches the decision boundary. In other words, the additive stage occurs *after* a decision has been made but before the RT is detected. In standard sequential sampling models, this type of post-decision/pre-report interval is referred to as terminal non-decision time (NDT; *Ratcliff and McKoon, 2008*; *Resulaj et al., 2009*). As noted above, we estimated that in our data, this transition occurs approximately 100ms prior to the RT. Because the focus of this study was to characterize the influence of gaze on the decision process itself (i.e. the putative first stage), the remaining analyses were conducted using data that were truncated by 100ms, in order to minimize the influence of post-decision gaze effects. However, as this is merely an estimate of the NDT interval, we repeated each analysis using data truncated over a range of NDT values from 0 to 400ms. The results, presented in *Figure 6—figure supplement 1*, and *Figure 7—figure*

*supplement 1*, indicate that the gaze biases described in the following sections are robust to NDT truncation values of up to 200ms.

## Initial fixation bias

As is the case in humans, the monkeys were more likely to choose the target that they looked at first in a given trial. To quantify this bias, we used a logit mixed effects regression, which gives an estimate of the effect of an initial leftward fixation on the log odds of a leftward choice: $\beta_{first-is-left}$=0.66, CI = [0.32; 1.00]; F(1,30043)=14.36, p=0.00015, see *Supplementary file 1*, 13. The fact that $\beta_{first-is-left}$ is significantly greater than zero indicates that an initial leftward fixation is associated with a greater likelihood of a leftward choice. The effect of target value differences computed in this same model was $\beta_{value-difference}$ = 1.40 (CI = [1.32, 1.43]; F (1,30043)=5349.40, p<1e-10). The ratio between these two regression estimates ($\beta_{first-is-left}$ / $\beta_{value-difference}$), gives the relative influence the initial fixation on choice as compared to the value differences of the targets: 0.66/1.40=0.47. In other words, the monkeys chose as if the first-fixated item was, on average, worth 0.47 drops of juice more than its nominal value. Critically, because the targets were obscured until the monkeys initiated their first saccade (see *Methods*), their initial fixations were unbiased with regard to target value. Thus, the choice bias in favor of the initially fixated item cannot be attributed to a tendency for the monkeys to look at higher value targets first.

This suggests that the initial fixation direction, whether exogenously or endogenously driven, may have a causal influence in choice. To test this causal role, we conducted additional experimental sessions in which we manipulated initial fixation by staggering the onset of the two targets (Monkey C: 16 sessions, n=6931 trials; Monkey K: 8 sessions, n=5433 trials). Staggered target onsets – either left first or right first – occurred randomly in 30% of the trials in these sessions (*Figure 3A*), with concurrent onsets in the other 70% of trials (*Figure 3B*). In the staggered-onset trials, the monkeys almost always directed their initial fixations to the target that appeared first (100% of staggered trials for Monkey C, 94.1% for Monkey K). Trials in which Monkey K did not fixate on the first target (n=99) were removed from analysis. As in the sessions without gaze manipulation described above, we found a main effect of initial fixation on choice (logit mixed effects regression: $\beta_{first-is-left}$=0.70, CI = [0.41; 1.00]; F(1,12359)=21.83, p=3.02e-06; $\beta_{first-is-left}$ / $\beta_{value-difference}$ = 0.57 drops of juice; see *Figure 5* and *Supplementary file 1*, row 14). Importantly, we found no main effect of trial-type (staggered vs. standard; $\beta_{trial-type}$ = -0.07, CI = [–0.33; 0.18]; F(1,12359)=0.32, p=0.57) nor was there a difference in the magnitude of the initial fixation effect across the two trial-types $\beta_{first-is-left:trial-type}$ = 1.48e-3, CI = [–0.23; 0.24]; F(1,12359)=1.52e-4, p=0.57. Thus, not only could we induce an initial fixation bias by manipulating where the monkey looked first, but the magnitude of this bias is comparable to the bias observed without a manipulation.

## Cumulative gaze-time bias

Here, we asked whether monkeys were more likely to choose items that were fixated longer in a trial. We first quantified the time spent looking at the two choice targets in every trial. Then, using a logit mixed effects regression, we asked whether choice outcomes were dependent on the relative time spent looking at the left item as compared to the right item ('Time Advantage Left'). Consistent with human choice patterns, monkeys were more likely to choose the left item as the relative time advantage for the left item increased (*Figure 6B and C*; $\beta_{time-advantage}$ = 18.71, CI = [15.07; 22.34]; F(1,28393)=101.76, p<1 e- 10, *Supplementary file 1*, row 15). Because fixation durations are themselves dependent upon the target values (see *Fixation Durations Depend on Target Value*, above), we repeated this analysis using choice probabilities that were corrected to account for the target values (see *Methods*). Using these corrected choice probabilities, the positive relationship between relative viewing times and choice was maintained. (*Figure 6D and C*; linear mixed effects regression: $\beta_{time-advantage}$ = 1.07, CI = [0.92; 0.1.21]; F(1, 28393)=208.90, p<1 e- 10, *Supplementary file 1*, row 16). Intuitively, this effect size would predict a change in choice probability of ~5% given a change in relative viewing time advantage from –0.1s to 0.1s. Thus, the monkeys were more likely to choose items that were fixated longer, independent of the values of the targets in each trial. The corrected cumulative gaze time bias effects were significantly above zero for data truncated up to 200ms (*Figure 6—figure supplement 1*).

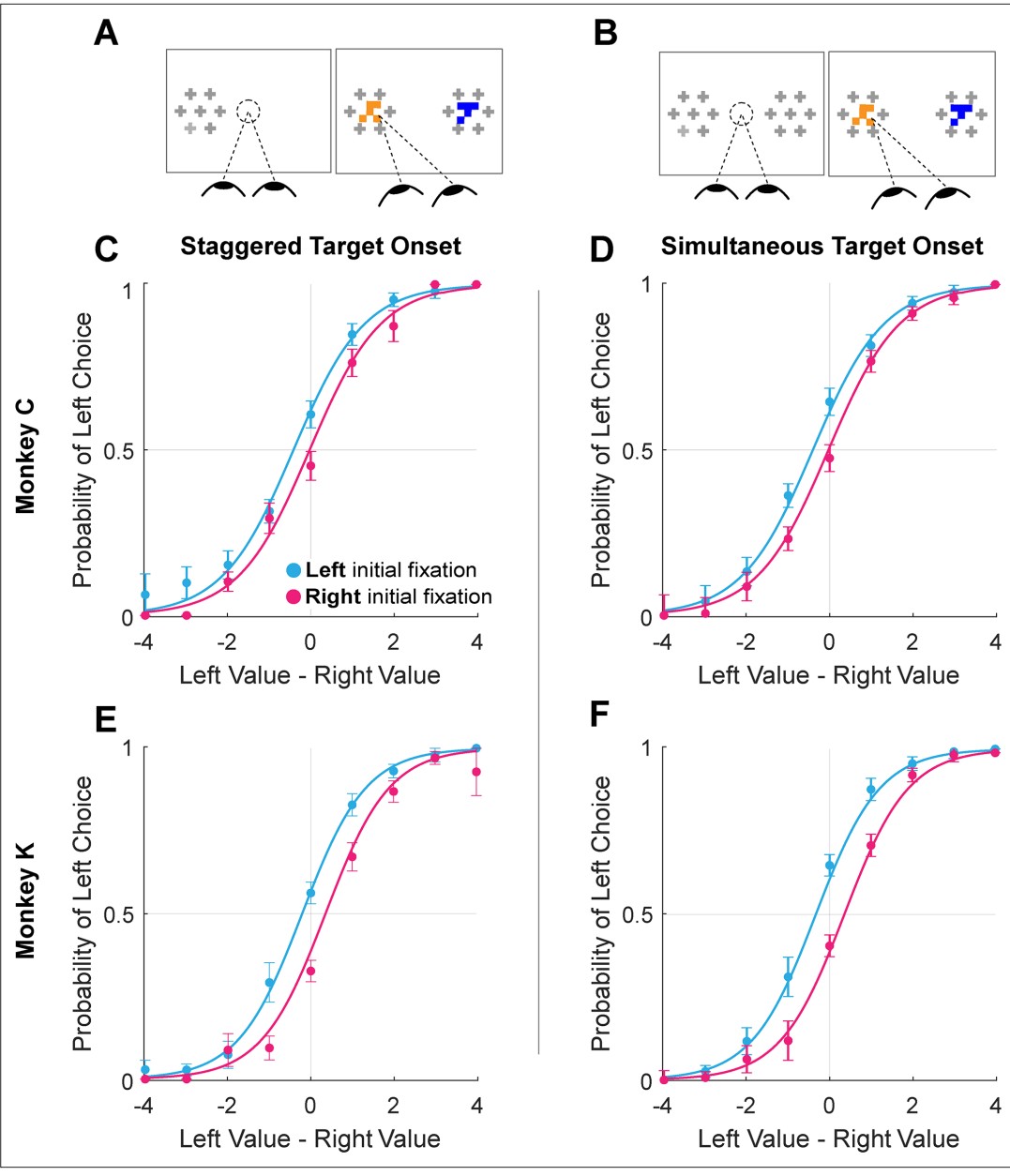

**Figure 5.** Initial gaze bias for gaze-manipulation sessions. (**A**) Depiction of gaze manipulation procedure. In 30% of trials, only a single target was initially presented, randomly assigned to the left or right of the display (left panel). Once a saccade was detected (or 250ms elapsed, whichever happened first), the second target appeared (right panel). (**B**) Depiction of standard trials with masked targets appearing simultaneously on both the left *and* right of the display (left panel). Masks disappeared when the initial saccade was detected (right panel). (**C–F**) Probability of choosing left as a function of the difference between the value of the item on the left and the value of the item on the right. Data were split according to the direction of the initial fixation (cyan and magenta) Circles show the mean probabilities of left choice, error bars show the standard error of the mean over 16 sessions for Monkey C and 8 sessions for Monkey K. Lines show logistic fits from the mixed effects model (*Supplementary file 1*, row 13). (**C & E**) show data from trials where the onset of the first target was staggered (N = 3449). (**D & F**) show trials where the targets appeared simultaneously (N = 8915). (**C–D**) shows data from Monkey C; (**E–F**) shows data from Monkey K.

## Final fixation bias

A prediction of earlier aSSM models is that the item being fixated at the end of the trial should be chosen more often than the non-fixated item, due to the net increase in evidence accumulation for fixated items, and consequently the greater likelihood that evidence for a fixated item reaches a

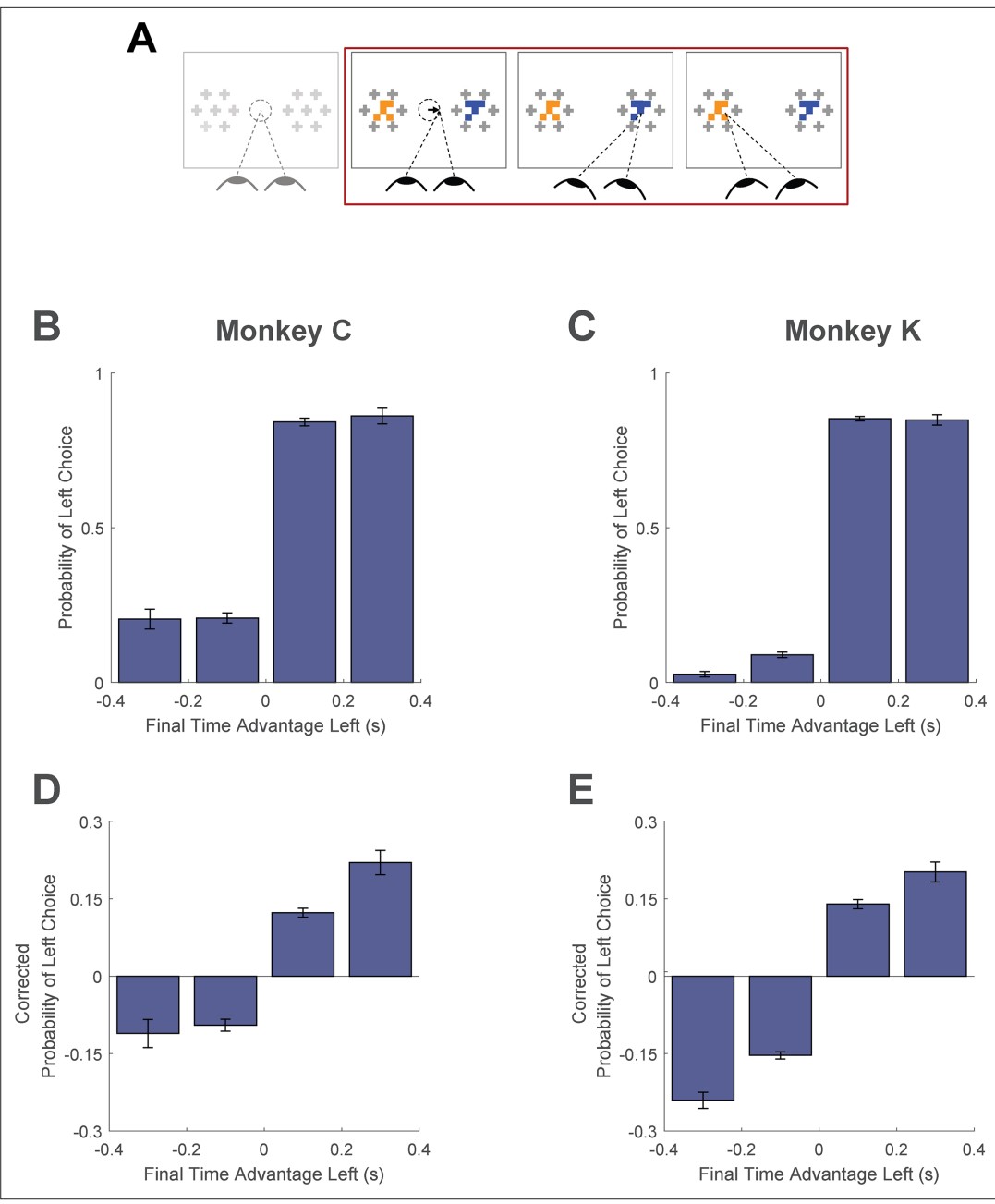

**Figure 6.** Cumulative gaze-time bias. (**A**) Task schematic. The portion of the trial used to show the cumulative gaze-time bias is highlighted in red. (**B–C**) Probability of choosing left as a function of the binned final gaze-time advantage for the left item. (**D–E**) The same as B-C, but using choice probabilities corrected to account for the target values in each trial (see *Methods*). For visualization, these graphs exclude those trials with a final time advantage greater than +/-3 standard deviations from the mean (<1% of trials for each monkey). The trials were not excluded from the logit mixed effects regression reported in the text. In B-E, the bin boundaries were 0, ±0.2, and ±∞. Error bars indicate SEM across sessions. The effect of NDT truncation on this bias is shown in *Figure 6— figure supplement 1*. In this figure, beta values in A and B refer to the estimate for variable time-advantage in *Supplementary file 1*, rows 15 and 16 (respectively), with standard errors derived from the mixed-effects model. Panels B and D show data from Monkey C (N = 15,175 trials); panels C and E shows data from Monkey K (N = 13,220 trials).

The online version of this article includes the following figure supplement(s) for figure 6:

**Figure supplement 1.** The effect of NDT truncation on the cumulative gaze bias.

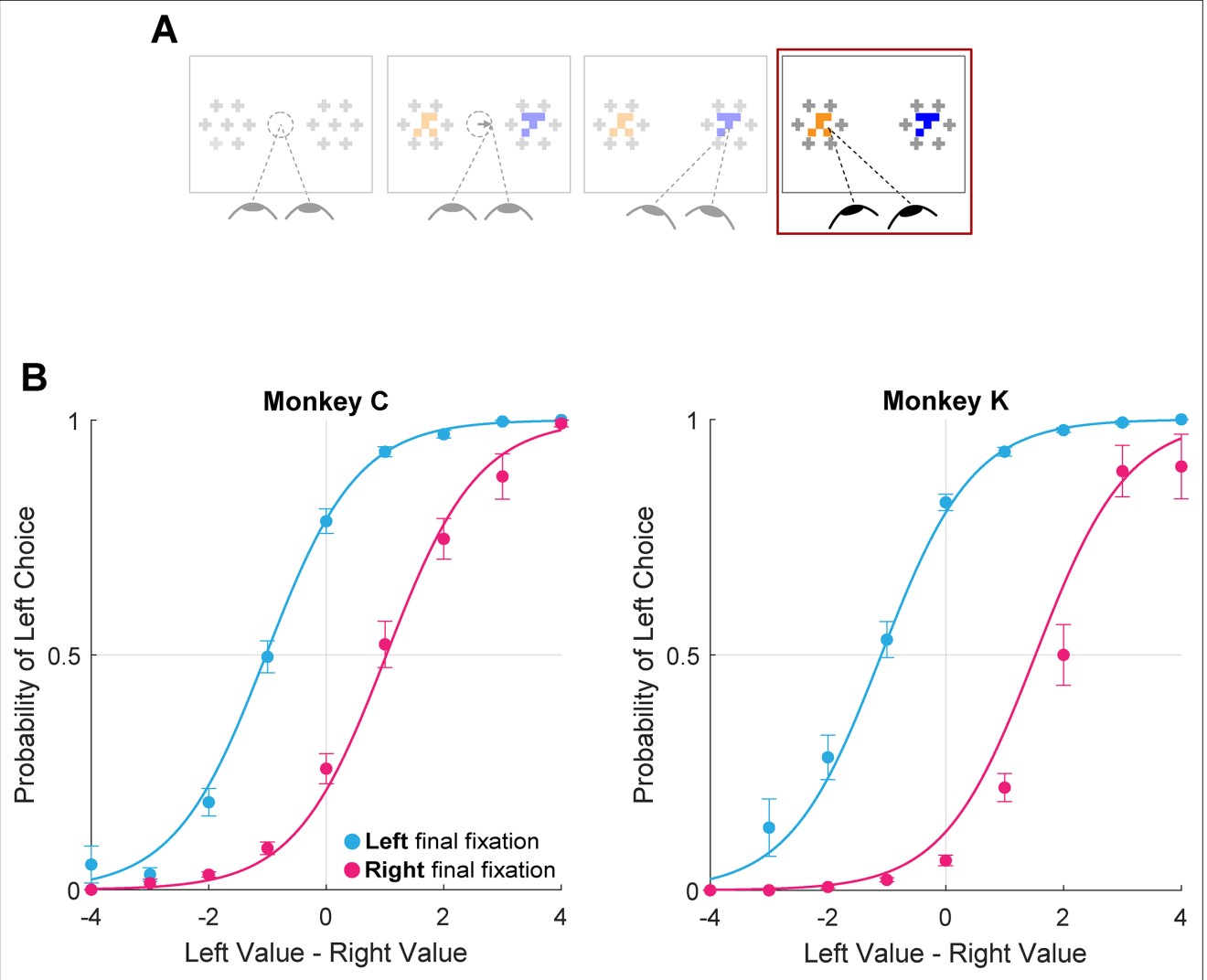

**Figure 7.** Final fixation bias. (**A**) Task schematic highlighting the portion of the trial used to show the final fixation bias (red box). (**B–C**) Probability of choosing left as a function of the difference in item values, split according to the location of gaze (left or right) at the end of the trial. Circles show mean probabilities of left choice, error bars show the standard error of the mean over 25 sessions for Monkey C and 29 sessions for Monkey K. Lines show logistic fits from the mixed effects model (***Supplementary file 1***, row 17). Panel B shows data from Monkey C (N = 15,175 trials); panel C shows data from Monkey K (N = 13,220 trials). The effect of NDT truncation on this bias is shown in ***Figure 7—figure supplement 1***. In this figure, beta value refers to the estimate for variable last-is-left ***Supplementary file 1***, row 17, with standard errors derived from the mixed-effects model.

The online version of this article includes the following figure supplement(s) for figure 7:

**Figure supplement 1.** The effect of NDT truncation on the final fixation bias.

bound at any given moment (***Krajbich et al., 2010***; ***Krajbich and Rangel, 2011***; ***Tavares et al., 2017***). In contrast, the two-stage model described by ***Westbrook et al., 2020***, suggests that the final fixation reflects, rather than influences, the result of the accumulation process. Therefore, we assessed the final fixation bias in our data in a manner similar to prior studies but with one important difference: as detailed above, the fixation data in each trial were truncated by 100ms to minimize the influence of late-stage gaze effects. In the truncated data, the location of the final fixation in each trial is defined as the target being viewed by the monkey at the *virtual* end of the trial, which in an aSSM framework can be interpreted as an estimate of when the decision boundary was reached (e.g. ***Resulaj et al., 2009***).

Using the truncated data, we found that the monkeys were more likely to choose the target being viewed at the virtual trial end, consistent with the final fixation bias effects observed in humans (***Figure 7***, logit mixed effects regression: $\beta_{\text{last-is-left}}=3.35$, CI = [2.34; 4.37]; $F(1,28392)=42.00$, $p<1e$-10, ***Supplementary file 1***, row 17). Note that this effect is present even after removing up to 200ms

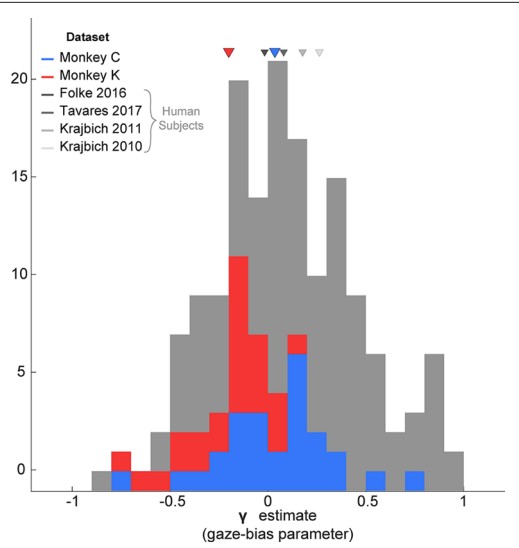

**Figure 8.** Distribution of gaze bias parameter (γ) across species. X-axis shows the range of possible values of the gaze bias parameter, γ. The gray distribution combines the γ values estimated for each subject in the four human datasets previously fit using the GLAM (*Thomas et al., 2019*). The means for each dataset are given as triangles above the distributions. The blue and red distributions show the session-wise distribution of γ parameters for Monkeys C, and K, respectively. Triangles in the corresponding colors show the means of the monkeys' distributions. Total number of subjects for each dataset are as follows: *Folke et al., 2016*: 24 subjects; *Tavares et al., 2017*: 25; *Krajbich and Rangel, 2011*: 30; *Krajbich et al., 2010* Total number of sessions for the monkeys were 29 and 25 for Monkeys C and K, respectively.

from the end of the trial (approximately 25% of the full trial duration, *Figure 7—figure supplement 1*). This means that final fixation biases can be explained at least in part by an effect of gaze on evidence accumulation – as suggested by prior studies (*Krajbich et al., 2010*)– when assuming a terminal NDT epoch of up to 200ms.

## Gaze-Weighted Linear Accumulator Model (GLAM)

The analyses above show broad consistency between NHP and human data with respect to the association between gaze and choice behavior. The question remains, however, as to whether these patterns emerge from a similar mechanism to those observed in humans. To this end, we asked whether an aSSM framework could explain the gaze and choice patterns we observed in NHPs. We chose the Gaze-weighted Linear Accumulator Model (GLAM; *Molter et al., 2019*), which was previously used to explain gaze-related choice biases in humans across multiple independent decision studies (*Thomas et al., 2019*). Using this model, we asked three questions with regards to our NHP data: First, does a model with a gaze bias mechanism fit our data better than a similar model without a gaze bias? Second, how do gaze biases estimated in our monkey subjects compare to prior human work? Third, can the model accurately predict the critical features of choice behavior, such as accuracy, RT, and the influence of gaze on choice?

To answer the first question, we fit two variants of the GLAM to each monkeys' data. These variants differed only in their treatment of the gaze bias parameter, γ (see *Equation 1* for model specification and parameter definitions). Note that since the model was fit using the NDT-truncated data, we exclusively utilized the multiplicative version of the model (See *Figure 4* and associated text). In the first GLAM variant, the gaze bias parameter γ was left as a free parameter, allowing gaze to modify the rate of evidence accumulation for fixated items. The second model fixed γ at 1, assuming a priori that gaze has no effect on the decision process. The maximum a posteriori (MAP) and 95% highest posterior density intervals (HPD) for the parameters estimated in both model variants are given in *Supplementary file 2*. The relative fit of the two variants was assessed using the difference in WAIC score (dWAIC, *Vehtari et al., 2017*), defined by subtracting the WAIC value for the model with γ fixed at 1 from the WAIC for the model where γ was a free parameter. The dWAIC for Monkey C was –3357.9, and for Monkey K was –8159.7, indicating that the model that allowed for the presence of gaze biases is the better fit of the two model variants. Further, this difference was significant as indicated by nonoverlapping WAIC values ±their standard error (*Supplementary file 1*). In the gaze-bias model, γ values were as follows: Monkey C: MAP = 0.079, HPD = [0.045, 0.12]; Monkey K: MAP = –0.056, HPD = [-0.087,–0.018].

Since the GLAM had previously been used to explore gaze biases in four human decision-making datasets (*Folke et al., 2016*; *Krajbich et al., 2010*; *Krajbich and Rangel, 2011*; *Tavares et al., 2017*), we were able to directly compare the extent of the gaze bias by comparing the γ parameters for humans vs. our monkey subjects. Treating each session as an independent sample, we found that the

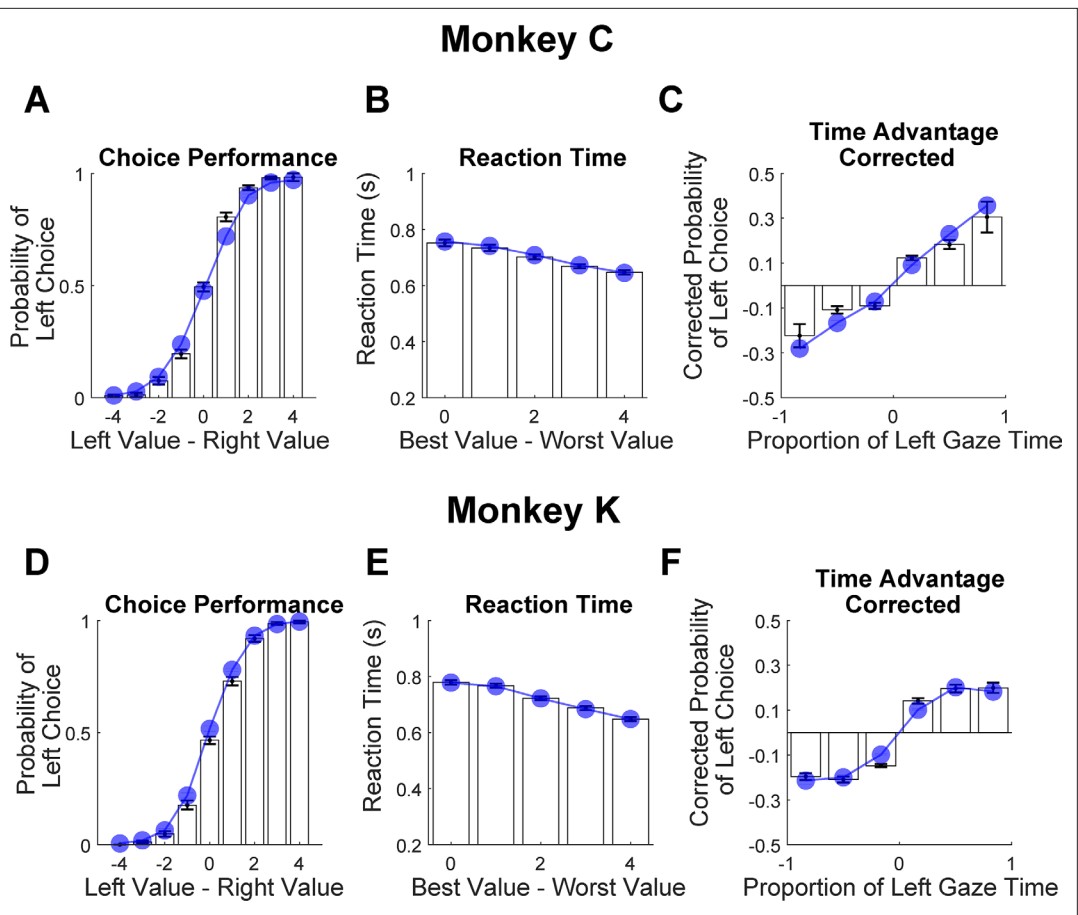

**Figure 9.** Out-of-sample predictions of the GLAM model for choice and gaze behavior. White bars show the empirically observed behavioral data from monkeys in odd-numbered trials. Blue dots and lines show GLAM model predictions for odd-numbered trials. Error bars show the standard error of the mean (SEM) across 25 sessions for Monkey C and 29 sessions for Monkey K. (**A,D**) Fraction of left choices as a function of the left minus right target value (in units of juice drops). (**B,E**) Reaction time decreases as a function of difficulty; note that reaction times are shorter here than in *Figure 1E*, due to the use of NDT-truncated data. (**C,F**) Corrected probability of choosing the left target as a function of the fraction of time spent looking at the left target minus fraction of time spent looking at the right target. The bin boundaries are 0,±0.33,±0.67, and ±1. Panels A-C correspond Monkey C (N = 15,612 trials); panels D-F correspond to Monkey K (N = 14,433 trials).

distribution of γ in each of our monkey subjects fully overlapped with the γ distributions from human studies (*Figure 8*).

Finally, we assessed the model's predictive accuracy by performing out-of-sample simulations to predict core features of the behavioral data. To do this, the data were first split into even- and odd-numbered trials. Next, the models were estimated using only the even-numbered trials. Finally, these estimates were used to simulate each of the held-out trials 10 times, resulting in a set of simulated choice and reaction time measures that could be compared to the empirically observed behavior in the held out trials. As illustrated in *Figure 9*, the model predictions closely match the monkeys' actual performance (see also *Supplementary file 1*, rows 18–20 and *Supplementary file 4*). Note that repeating this procedure with γ fixed at 1 (no gaze bias) produces accurate predictions of choices and reaction times but fails to capture the effect of cumulative gaze-bias on choice (not illustrated, see *Thomas et al., 2019*).

In summary, an aSSM model that includes gaze bias provides a better fit than a model without a gaze bias, and the magnitude of this bias is comparable to that estimated in human data using the same modeling framework. Finally, the model is able to make accurate out-of-sample predictions of choice and gaze behavior.

# Discussion

Over the last decade, there has been a growing appreciation for the active role that attention plays in a broad array of decision-making domains. However, the neural substrates of this relationship remain largely unknown due to the lack of a suitable animal model. The aim of the present study was to develop such a model using a novel value-based decision-making task for NHPs. We validated our animal model on two measures: behavioral control and the presence of human-comparable gaze biases. First, the monkeys' performance indicated good behavioral control: both choices and RTs were a graded function of task difficulty (defined by the difference in target values), with fast and nearly optimal choices in the easiest trials, and slower and more variable choices in harder trials. Such signatures of behavioral control are crucial to the development of any animal model of complex cognitive processes, as they allow the confident attribution of an animal's behavior to the construct of interest, and not to attentional lapses or guesses (*Fetsch, 2016*; *Krakauer et al., 2017*). For the present study, this suggests that the monkeys' behavior is ascribable to a decision process driven primarily by the values of the targets in each trial. As to the second criterion, our results show that the monkeys exhibit similar gaze biases to those observed humans, both in terms of the core behavioral measures and also in their ability to be mechanistically described using an aSSM. Taken together, we conclude that this novel task is well suited for use in neural mechanistic studies, such as concurrent electrophysiological recordings or stimulation.

## Comparison to prior human studies

While our findings are broadly consistent with those from humans, there are two notable differences between the monkeys' gaze behavior in the present study, and that of humans reported in prior studies. First, monkeys made fewer fixations per trial than humans on average (~2.5 vs 3.7); second, the monkeys' final fixations in each trial tended to be prolonged compared to non-final fixations—the opposite of what has been reported in many human studies. Critically, these two differences may be related. For both humans and monkeys, fixations made early in the trial are longer than those made later; thus, because the monkeys' final fixations occur early in the trial (e.g. either 2nd or 3rd), they will be longer on average than final fixations in humans, which tend to occur later in the trial (e.g. 4th, 5th, etc.).

One significant similarity between the current study and prior studies in humans was the positive association between gaze and choices, which was well-explained using an aSSM. As noted in the results, our data are most consistent with a two-stage aSSM in which gaze initially biases the choice process, and then (after a decision boundary is reached) transitions to a mode in which gaze becomes drawn to the intended choice target (*Manohar and Husain, 2013*; *Westbrook et al., 2020*). The behavioral feature of this second stage is consistent with a motor-preparatory phenomenon known as a gaze-anchoring, in which gaze becomes 'locked' onto the reach target in order to increase the speed and efficiency of the response (*Battaglia-Mayer et al., 2001*; *Dean et al., 2011*; *Neggers and Bekkering, 2000*).

In prior studies of gaze-anchoring in humans and NHPs, the latency between the eyes locking onto target of an upcoming response and the beginning of detected movement ranges between 64 and 100ms (*Battaglia-Mayer et al., 2001*; *Neggers and Bekkering, 2001*; *Reyes-Puerta et al., 2010*; *Stuphorn et al., 2000*). The terminal NDT estimate that we derived from the model comparison is consistent with these latencies, and with NDT estimates obtained from standard sequential sampling models (e.g. *Resulaj et al., 2009*). Ultimately, the true duration of the post-decision epoch is difficult to estimate from behavioral data alone. Additional work will be necessary to either measure or infer the plausible range of post-decision/pre-report intervals in our task, in order to confirm the robustness of the gaze bias effects measured here.

## Comparison to prior NHP studies

Beyond similarities to prior human studies, our results are also consistent with the handful of NHP studies in which some degree of free eye movement was permitted. Among these, both *Hunt et al., 2018* and *Rich and Wallis, 2016* imposed some restrictions on eye movements, making it difficult to assess the relationship between fully natural gaze patterns and choices. *Cavanagh et al., 2019* used a fully free-viewing design, and also observed a bias in favor of initially fixated items.

Our study confirms this result, and provides additional insights, due to novel elements of the study design. First, by manipulating initial fixation direction, we show that initial fixation has a causal effect on the decision process. Second, the results of the aSSM model-fitting provide a mechanistic explanation for our results and permits a direct comparison to model fitting results in human studies.

Finally, unlike many prior studies, the decision task requires the monkeys to sample the targets by fixating them directly, due to the use of a gaze contingent mask that completely obscures the stimuli at the beginning of the trial, and the use of visual crowders that closely surround each target. This subtle design element is crucial, because it permits accurate measurement of the time spent attending to each target, and of the relationship between relative viewing time and choice. In a forthcoming study using this same task, we show that OFC neurons do not begin to reflect the value of the second-fixated stimulus until well after it has been viewed by the monkey, indicating that value information is only available to the monkey once a target is fixated (*McGinty and Lupkin, 2021*). In contrast, in tasks where value-associated targets are easily perceived in peripheral vision, even targets that are not fixated influence prefrontal value signals and decision behavior (e.g. *Cavanagh et al., 2019*; *Xie et al., 2018*). High-value peripheral targets are also implicated in value-based attentional capture, in which overt or covert attention becomes drawn to objects that are reliably associated with reward (*Anderson et al., 2011*; *Kim et al., 2015*). Thus, the task used in the present study minimizes value-based capture effects.

## Hypothesized neural substrates

The impetus for developing this animal model is to explore the neural underpinnings of the relationship between gaze and value-based choice. Given the fact that a sequential sampling model provides objectively good fits to the observed behavior, one potential approach for understanding neural mechanisms is to identify brain regions that correspond to different functions in a sequential sampling framework. Such an approach has been used in perceptual dot motion discrimination, where extrastriate cortical area MT is thought to provide the 'input' signal (encoding of visual motion) and neurons in parietal area LIP are thought to represent the accumulated evidence over time (see *Gold and Shadlen, 2007*, for review). One note of caution, however, is that brain activity that outwardly appears to implement accumulator-like functions may not be causally involved in the choice (e.g. *Katz et al., 2016*). Nonetheless, it is still useful to consider the neural origins of two core computations suggested by sequential sampling models: the representation of the target values (corresponding to the input signals), and neural signals that predict of the actions performed to obtain the reward (reflecting the decision output).

For the encoding of accumulated evidence, findings from NHP neurophysiology point to regions involved in the preparation of movement. In motion discrimination tasks that use eye movements to report decisions, accumulated evidence signals can be observed in oculomotor control regions such as area LIP, frontal eye fields, and the superior colliculus. Because the decision in our task is reported with a reach movement, potential sites for accumulator-like activity include motor cortical areas such as the arm representations within dorsal pre-motor cortex (*Chandrasekaran et al., 2017*). In addition, human imaging studies have identified several other candidate regions with accumulator-like activity, including in the intraparietal sulcus, insula, caudate, and lateral prefrontal cortex (*Gluth et al., 2012*; *Hare et al., 2011*; *Pisauro et al., 2017*; *Rodriguez et al., 2015*). Interestingly, several studies find accumulator-like signals in cortical areas in the dorsal and medial frontal cortex, but these are not consistently localized. For example, whereas *Pisauro et al., 2017* find accumulator-like activity in the supplementary motor area, *Rodriguez et al., 2015* identify a more anterior region in the medial frontal cortex. Additional studies in both humans and NHPs will be necessary to understand the specializations of these regions with respect to encoding decision evidence.

Because decision behavior in this study was value-dependent, regions encoding economic value are strong candidates for signals that provide input to an accumulator. Value-related signals have been observed in numerous cortical and subcortical regions, including the amygdala (e.g. *Paton et al., 2006*), the ventral striatum (*Kable and Glimcher, 2009*; *Lim et al., 2011*), and the ventromedial frontal lobe (VMF), including both the orbitofrontal and ventromedial prefrontal cortices (*Padoa-Schioppa and Assad, 2006*; *Rich and Wallis, 2016*; *Vaidya and Fellows, 2015*). Of these regions, the VMF has received the most scrutiny with regard to gaze-biases. In humans, there is evidence of BOLD activity in this region related to attention-guided decision-making (*Gluth et al., 2012*; *Hare et al., 2011*; *Leong*

*et al., 2017*; *Lim et al., 2011*; *Pisauro et al., 2017*; *Rodriguez et al., 2015*), and patients with lesions to this area show impairments in such tasks (*Vaidya and Fellows, 2015*).

In NHPs, the orbitofrontal portion of the VMF (the OFC) has been demonstrated to contain value-signals that are dependent on whether gaze is directed towards a reward-associated visual target (*Hunt et al., 2018*; *McGinty et al., 2016*; *McGinty and Lupkin, 2021*) a mechanism fully consistent with the effects of gaze posited by aSSM frameworks. However, the precise neural mechanisms explaining how gaze biases choice outcomes are still unknown. One hypothesis is that the value-coding regions inherit gaze-modulated information from visual cortical regions sensitive to shifts in visual attention. For instance, neurons in the anterior inferotemporal cortex, which projects directly to the OFC (*Carmichael and Price, 1995*; *Kravitz et al., 2013*), selectively encode features of attended objects to the exclusion of others in the same receptive field (*DiCarlo and Maunsell, 2000*; *Moore and Armstrong, 2003*; *Moran and Desimone, 1985*; *Richmond et al., 1983*; *Sheinberg and Logothetis, 2001*). Therefore, the similar fixation-driven effects on OFC value-coding shown by McGinty and colleagues (*McGinty et al., 2016*; *McGinty and Lupkin, 2021*) may be inherited from this structure. Another candidate region for the top-down influences on gaze-dependent value-coding, is the frontal eye fields (FEF). In a recent study from *Krajbich et al., 2021*, focal disruption of FEF leads to a reduction in the magnitude of gaze-biases. Prior work in both humans and NHPs has shown that the FEF can influence perceptual sensitivity through the top-down modulation of early visual areas (*Moore and Armstrong, 2003*; *Moore and Fallah, 2004*; *Ruff et al., 2006*; *Silvanto et al., 2006*; *Taylor et al., 2007*), suggesting that it may exert similar influence on value-encoding regions.

## Conclusions

In real-world decision scenarios, both humans and NHPs depend heavily on active visual exploration to sample information in the environment and make optimal decisions. In this way, even simple decisions require dynamic sensory-motor coordination. The importance of these gaze dynamics is underscored by the growing number of human studies showing that gaze systematically biases the decision process. We have developed a novel value-based decision-making task for macaque monkeys and found that monkeys spontaneously exhibit gaze biases that are quantitatively similar to humans', while also showing the high level of behavioral control necessary for neurophysiological study. This novel paradigm will therefore make it possible to identify the cell- and circuit-level mechanisms of these biases.

## Methods
### Subjects

The subjects were two adult male rhesus monkeys (*Macaca Mulatta*), referred to as Monkey C and Monkey K. Both subjects weighed between 13.5 and 15 kg during data acquisition, and both were 11 years old at the start of the experiment. They were implanted with an orthopedic head restraint device under full surgical anesthesia using aseptic techniques and instruments, with analgesics and antibiotics given pre-, intra-, and post-operatively as appropriate. Data from Monkey K were collected at Stanford University (25 sessions, 14,433 trials); data from Monkey C were collected at Rutgers University—Newark (29 sessions, 15,613 trials). All procedures were in accordance with the Guide for the Care and Use of Laboratory Animals (*National Research Council, 2011*) and were approved by the Institutional Animal Care and Use Committees of both Stanford University and Rutgers University—Newark.

### Study design and apparatus

The subjects performed the task while head-restrained and seated in front of a fronto-parallel CRT monitor (120 Hz refresh rate, 1024x768 resolution), placed approximately 57 cm from the subjects' eyes. Eye position was monitored at 250 Hz using a non-invasive infrared eye-tracker (Eyelink CL, SR Research, Mississauga, Canada). Three response levers (ENV- 612 M, Med Associates, Inc, St. Albans, VT) were placed in front of the subjects, within their reach. The 'center' lever was located approximately 17 cm below and 35 cm in front of the display center, and the center points of the two other levers were approximately 9 cm to the left and right of the center lever. The behavioral paradigm was run in Matlab (Mathworks, Inc, Natick, MA) using custom scripts using the Psychophysics Toolbox

extensions (*Brainard, 1997*; *Cornelissen et al., 2002*) . Juice rewards were delivered through a sipper tube placed near the subjects' mouths.

## Behavioral task

### Task structure

*Figure 1A* shows an abbreviated task sequence; *Figure 1—figure supplement 1A* shows the full sequence as described here: Each trial began with a fixation point in the center of the screen. Upon appearance of the fixation point, the subjects were required to both depress the center lever and to maintain fixation within a radius of 3.5 degrees of visual angle from the center of the fixation point. The monkeys were required to use only one hand to perform the task (Monkey K used his right hand; Monkey C used his left). After the initial fixation and center lever press were both held for an interval between 1 and 1.5 s, the fixation point disappeared, and two target arrays were presented 7.5 degrees to the left and right of the fixation point. Each target array consisted of a choice target, an initial mask, and six non-relevant 'visual crowders'. The net effect of the mask and crowders was to obscure the target until the monkey looked directly at the target with a saccadic eye movement. See *Gaze Contingent Mask and Visual Crowders,* below, for details. The two targets shown in each trial were selected randomly without replacement from a set of 12 (see *Figure 1B* & *Figure 1—figure supplement 1C*) and placed randomly on the left or right side of the display, giving 132 unique trial conditions.

Once the target arrays appeared, the monkeys were free to move their gaze as they desired and were free to initiate a choice at any time. To initiate a choice, the monkeys lifted their hand off the center lever, at which point the target arrays were extinguished. After releasing the center lever, the monkeys had 500 or 400ms (for Monkey C and K, respectively) to choose their desired target by pressing the left or right lever. After a delay (1 s for Monkey K; 1 or 1.5 s for Monkey C), the subjects received the volume of juice reward associated with the chosen target (range 1–5 drops). If neither the left nor right lever was pressed within 400 or 500ms of the center lever lift, the trial ended in an error. Each trial was followed by an inter-trial interval randomly drawn from a uniform distribution between 2 and 4 s. Eye movements following the initial fixation were monitored but not enforced, and had no programmatic influence on the outcome of the trial. Reaction time (RT) was calculated as the time between target arrays onset and the release of the center lever.

### Choice targets

Choice targets were constructed as composites of two features, color and shape, and the reward value of a given target (range 1–5 drops) was determined by the sum of the reward values associated with its two features. (*Figure 1—figure supplement 1C*). Notably, this scheme allows for multiple visually distinct targets to share the same value. An example target set is shown in *Figure 1B* and *Figure 1—figure supplement 1C*. For a given set, the colors were selected by randomly sampling four equally-spaced hues from a standardized color wheel based on the CIELUV color space, and shapes were quasi-randomly selected from a library of 28 equal-area shapes.

New stimulus sets were generated frequently to avoid over-training on a given stimulus set. For every new set, 1–2 training sessions (not used for analysis) were performed so that the monkeys could become familiar with the stimuli, followed by 1–3 sessions of data collection.

### Gaze-contingent initial mask and visual crowders

A key design goal of this task was to encourage the monkeys to fixate onto the choice targets directly using saccadic eye movements. To encourage this behavior, we used two methods to obscure the choice targets until they were fixated directly.

The first method was a gaze-contingent initial mask: when the target arrays first appeared in each trial (i.e. the first frame), the choice targets were not shown; rather, in their place were two non-relevant, non-informative masking stimuli-drawn from a pool of visual crowder stimuli (described below). The masking stimuli remained until the monkeys' moved their eyes out of the initial fixation window by initiating a saccade, usually ~150ms after the onset of the arrays. Once the eyes left the initial window, the masking stimuli were replaced with the actual choice targets in the next display frame. Because the display frame rate (one frame every 8.25ms) was shorter than the typical time taken to complete a saccade (~30–40ms), the switch from the masks to the actual stimuli occurred

while the monkeys' eyes were moving, so that the choice targets appeared on the display before the end of the saccade (See *Figure 1—figure supplement 1A* for a step-by-step illustration). Therefore, information about target values was only available after the monkeys initiated a saccade towards one of the two target arrays.

The second method was the use of 6 visual crowders placed in close proximity in a hexagonal arrangement around each target (*Figure 1C*). These crowders minimized the monkeys' ability to use peripheral vision to identify the choice targets (*Crowder and Olson, 2015*; *Whitney and Levi, 2011*), further encouraging them to make fixations directly onto the targets.

The initial mask stimuli and the visual crowders were constructed using plus (+) or cross (x) shapes, colored using five hues randomly sampled from equally spaced points on a standardized color wheel. These stimuli convey no value information because the plus and cross shapes were never used to construct target stimulus sets, and because the use of multiple, random hues within each of the stimuli does not signify any one color that could indicate value.

The crowders and initial target masking were highly effective at encouraging the monkeys to fixate both targets before initiating a choice: both targets were viewed in 97.2% of trials for Monkey C and 92.6% of trials for Monkey K. These methods also ensured that the first fixation following target onset was unbiased with respect to the target values: in trials where one target had a larger value than the other, the monkeys directed their first saccade towards the higher value target 49.9% of the time, which was not significantly different from chance ($t (25559)=–0.45, p=0.65$).

## Gaze manipulation

To test the causal nature of the relationship between initial fixation and choice, we conducted additional sessions in which we temporally staggered the onset of the two targets (Monkey C: 16 sessions, n=6931 trials; Monkey K: 8 sessions, n=5433 trials). During these sessions, in 30% of the trials the first target array would appear randomly on either the left or right of the screen. The second target array appeared 250ms after the onset of the first, or when the monkeys began their initial saccade away from the central fixation point, whichever came first. The location of the first target array was randomized across trials. This manipulation was intended to encourage the monkeys to direct their attention towards the target that appeared first, as shown in prior human studies (*Tavares et al., 2017*). As expected, the monkeys directed their initial fixation to the cued target on 96% of the staggered-onset trials.

## Behavioral analysis

With the exception of the computational modeling, analyses were performed in Matlab v. R2022a (Mathworks, Inc, Natick, MA). All graphs, except *Figure 3*, were generated within the Matlab environment. For *Figure 3*, the violin plots were generated using the Seaborn v. 0.12.2 library for Python (*Waskom, 2021*). Unless otherwise noted, analyses were conducted using generalized mixed-effects regressions with random effects for monkey-specific slopes and intercepts. To fit these models, data were collapsed across sessions, unless otherwise noted. Full specifications for all models are given in *Supplementary file 1*. Regression estimates (β's) and 95% confidence intervals (CI) are reported for each effect. Significance for each effect was determined using a post-hoc ANOVA in which the regression estimates were compared to 0 (*F*- and p- values, reported in the text).

The onset and duration of fixations, defined as periods of stable gaze upon one of the two targets, were recorded by the Eyelink CL software. For analysis purposes, a fixation was considered to be on a given target if it was located within 3–5 degrees of the target center. Two consecutive fixations made onto the same target were merged into one continuous fixation. Fixations that were not onto either target were discarded; however, these were extremely rare: in the vast majority of trials, the monkeys fixated exclusively upon the targets.

For analysis purposes, fixations were designated as 'first', 'final' or 'middle'. First fixations were defined as the first target that was fixated after the onset of the target arrays; final fixations were defined as the target being fixated at the moment the monkey initiates a choice by lifting the center lever; and middle fixations were defined as all on-target fixations that were neither first nor final. Fixation durations were measured in the conventional manner, except for final fixations, which were considered to be terminated at the time of choice in each trial, consistent with previous human studies (*Krajbich et al., 2010*; *Krajbich and Rangel, 2011*; *Tavares et al., 2017*; *Westbrook et al., 2020*).

When using truncated data (see *Estimation and Truncation of Terminal Non-Decision Time (NDT)*, *below*) final fixations were terminated at the virtual trial end point.

For analyses of the final fixation bias and the cumulative gaze-time bias, trials containing only a single fixation were excluded (Monkey C: N=447 trials (2.8% of total), Monkey K: N=1217 trials (8.4% of total)), as they could artificially inflate either or both of these biases. For the latter analysis, we computed in every trial the time advantage for the left item by subtracting the total duration of fixations onto the right target from the total duration of fixations onto the left target (*Krajbich et al., 2010*). Thus, for trials where the monkey allocated more gaze time to the left target, the time advantage takes a positive value, whereas it takes a negative value for trials where the monkey allocated more gaze time to the right.

Finally, for the cumulative gaze bias analyses shown in *Figure 6D and E*, choice proportions were corrected using the same procedure set forth in *Krajbich et al., 2010*. For each trial, we coded the choice outcome as 0 for a right choice and 1 for a left choice. Then from each trial's choice outcome we subtracted the average probability of choosing left for all trials in that same condition, where the condition was defined as the left minus right target values. This correction removes from the choice data any variance attributable to the differences in the target values; as a result, *Figure 4C and E* reflect the effect of cumulative gaze after accounting for the influence of the target values on gaze durations.

## Computational model

To gain insight into potential mechanisms underlying choice and gaze behavior in this task, we employed the Gaze Weighted Linear Accumulator Model (GLAM, *Thomas et al., 2019*). The GLAM modeling was conducted in Python version 3.7 using the *glambox* toolbox (see *Code Availability* for details). The GLAM is an aSSM characterized by a series of parallel accumulators racing to a single bound. These accumulators are co-dependent, such that their combined probability of reaching the bound is equal to 1. For two choice options, this formulation is mathematically equivalent to a single accumulator drifting between two bounds (*Thomas et al., 2019*; *Krajbich and Rangel, 2011*). The effect of gaze on choice behaviors is implemented as follows: The GLAM posits that over the course of a trial the absolute evidence signal for each item, *i*, can have two states: a biased state during periods when an item is viewed, and an unbiased state when it is not. The average absolute evidence for each item, $A_i$, is a linear combination of these two states throughout the trial. As noted in the results, there are two methods through which gaze could bias choice behavior: a multiplicative manner in which gaze amplifies the value of the attended target, and an additive manner in which gaze adds a fixed increase in evidence accumulation to the attended target, independent of its value.

Assuming a multiplicative bias, $A_i$ is defined according to *Equation 1a*.

$$A_i = g_i r_i + \left(1 - g_i\right) \gamma r_i \tag{1a}$$

Here, $r_i$ is the veridical value of target *i* (number of drops of juice), $g_i$ is the fraction of gaze time that the monkey devoted to that target during a given trial, and γ is the crucial gaze bias parameter, which determines the extent to which the non-fixated item is discounted. This parameter has an upper bound of 1, which corresponds to no gaze bias. A value of γ less than 1 indicates biased accumulation in favor of the fixated item.

Under an additive gaze-biasing mechanism, $A_i$ is defined according to *Equation 1b*.

$$A_i = r_i + \gamma g_i \tag{1b}$$

As in the multiplicative model, $r_i$ is the veridical value of target *i* (number of drops of juice), $g_i$ is the fraction of gaze time that the monkey devoted to that target during a given trial, and γ is the gaze bias parameter. However, in the additive model variant, a γ value of 0 corresponds to no gaze bias, and a nonzero γ value indicates biased accumulation in favor of the fixated item.

The gaze-adjusted absolute evidence signals for each item are then used as inputs to an evidence accumulation process, as follows: First, each absolute value signal ($A_i$) is converted to a relative evidence signal, $R_i$, through a logistic transform (see *Molter et al., 2019*; *Thomas et al., 2019* for details and rationale). Then, evidence for each item ($E_i$) at time *t* is accumulated according to *Equation 2*, where $v$ and σ indicate the speed of accumulation and the standard deviation of the noise,

respectively, and $R_i$ is the relative evidence signal. $N$ indicates a univariate normal distribution. A choice is made when the evidence for one of the items reaches the threshold.

$$E_i(t) = E_i(t-1) + vR_i + N(0, \sigma^2) ; E_i(0) = 0 \tag{2}$$

We fit several different variants of the GLAM model to each monkey's data. To evaluate a hybrid two-stage modeling framework, we compared the relative fit of multiplicative and additive models to the data (*Figure 4*). Given the overall better fit obtained from the multiplicative models, these were used for the remainder of the analyses. To evaluate the necessity of the gaze bias parameter γ for explaining behavior, we used two variants of the multiplicative model, one in which the gaze bias parameter was left as a free parameter, and another that assumed no gaze bias by setting $\gamma=1$ (results in reported in *Supplementary file 2*). Model fits were compared using the Widely Applicable Information Criterion (WAIC, *Vehtari et al., 2017*) which accounts for the differences in the complexity of the two model variants. Specifically, we used both the signed difference of WAIC values and the WAIC weights for each model, which can be interpreted as the probability that each model is true, given the data (*Vehtari et al., 2017*).

In addition, we assessed the model's predictive accuracy by estimating the parameters for the free- γ model variant using only even numbered trials, and then using these parameter estimates to simulate the results of each of the held-out odd-numbered trials 10 times. We quantified the goodness-of-fit of the simulations by comparing three outcome metrics: choice, RT, and corrected probability of a leftward choice as a function of the cumulative gaze time advantage for the left item (as in *Figure 4D and E*). Note that since value is removed from the corrected cumulative gaze effect metric (see above), the only input to the model that could modulate it was the relative amount of gaze time devoted to each target. Thus, we can use this metric to assess the model's ability to emulate the net effect of gaze alone on choice. For each metric we computed a mixed effects regression (logistic for choice, linear for RT and the corrected cumulative gaze effect), in which a vector including both the empirically observed data and the simulations was explained as a function of a binary variable indicating whether a data point came from the empirically observed data, or from the model simulations (i.e. *Metric ~1 + isSimulated*). If the fixed effect of this variable was not significantly different from 0, then the simulated data did not deviate from the empirically observed data. The mean deviations for each metric are given by the beta-weights and are reported in *Supplementary file 4* along with the confidence intervals. As with the other regression models used in this paper, 'Monkey' was included as both a random slope and intercept.

All GLAM parameter estimation procedures were identical to those set forth in *Thomas et al., 2019*, with one exception: Thomas and colleagues assumed a fixed rate of random choice of 5%. This was changed to 1% to reflect the extremely low lapse rate for the easiest trials shown by both monkeys. The lapse rate for each monkey was estimated by finding the percentage of trials in which the monkey chose a 1-drop target when a 5-drop target (the highest possible value) was also available (mean lapse rate: Monkey C=0.64%, Monkey K=0.33%).

## Estimation and truncation of terminal non-decision time (NDT)

In sequential sampling models, the interval between when a decision is reached and when it is detectable to the experimenter is referred to as non-decision time (NDT). To account for and minimize the influence of gaze behavior in this interval, we truncated the gaze data using an estimate of this post-decision epoch. To obtain this estimate, we leveraged the key assumption of the two-stage aSSM, namely that gaze acts in a multiplicative manner prior to a boundary crossing (i.e. reaching a latent decision) and then additively between the boundary crossing and when the decision is reported (i.e. the RT, *Manohar and Husain, 2013*; *Westbrook et al., 2020*) Therefore, by removing gaze data from the end of the trial, the probability of the data being better explained by an additive model (and not a multiplicative model) should decrease as more data are removed. We collapsed data across the two monkeys and fit both an additive and a multiplicative variant of the GLAM for data truncated at 0ms, 50ms, 100ms, 150ms, 200ms, 300ms, and 400ms. We then compared the posterior probabilities of the model being correct given the data and the alternative model using the *compare* function from PyMC. We also compared the WAIC for the two variants, for each level of truncation (*Supplementary*

*file 4*). The relative probability of the additive model generating the data decreases sharply between 0ms and 100ms of truncation, after which it reaches zero.

Accordingly, we selected 100ms as our NDT estimate, and for the data shown in the main figures we removed gaze data between center lever lift and 100ms before the lever lift in every trial. In other words, in every trial the gaze data were treated as if the trial had actually ended 100ms before the center lever lift; all calculations related to the gaze sequence and timing were calculated with respect to this virtual trial stopping point. For trials where the virtual stopping point occurred during an earlier fixation, we reassigned the final fixation location according to the location of the eyes at the virtual stopping point. In this way, the effects of late-trial gaze effects during the NDT were minimized. The truncated data were used to compute the gaze biases, except the initial fixation bias, and were also used as input to the computational model. Importantly, the main analyses were repeated using truncation values ranging from 0 to 400ms, with results shown in *Figure 6—figure supplement 1*, and *Figure 7—figure supplement 1*.

### Sample size determination

Because the behavioral task was novel, and because there were no published data from free-viewing NHP decision tasks at the time the study began, there were no effect size estimates available to facilitate a power analysis. We therefore began by collecting a pilot sample, and assessing the initial gaze bias and cumulative gaze bias effects as reported in humans by *Krajbich et al., 2010*. To minimize confirmatory bias stemming from exploratory analyses, these initial analyses procedures were identical to those in Krajbich et al. The first pilot sample, in Monkey K, consisted of 13 sessions (approximately 8000 trials). Using planned analyses from Krajbich et al., we identified statistically significant initial fixation and corrected cumulative gaze biases. Based upon this initial sample, we then targeted a per-monkey sample size of approximately 10–15 sessions or 7000–15,000 trials for detecting behavioral effects of interest in subsequent experiments.

We then collected additional sessions from Monkey K for the purposes of electrophysiological recordings. These were combined with previous behavior-only sessions to create the Monkey K data used in this study (total = 25 sessions). From the second monkey we first collected 13 behavior-only sessions, and then additional 16 sessions for the purpose of electrophysiological recordings, resulting in the 29 sessions for Monkey C reported here.

We defined outlier sessions as those in which the fraction of suboptimal decisions exceeded ~1% for the easiest choices (+/-3 or 4 on the X-axis in *Figure 1D*), indicating that the monkeys had not yet sufficiently learned the target values; these sessions were excluded from analysis. No exclusions were made on the basis of gaze data, or upon gaze-related outcome measures.

### Code Accessibility

The data and code in this folder are available at https://osf.io/hkgmn/. The *glambox* toolbox that was used for the computational modeling can be accessed at https://github.com/glamlab/glambox (*Molter et al., 2019*). The associated documentation can be accessed at https://glambox.readthe-docs.io/en/latest/.

## Acknowledgements

WT Newsome for funding, material support, and thoughtful discussions; A Rangel, F Molter, G Rosenbaum, G Karpov, E Murray, D Sharma, & A Thomas for thoughtful discussions and comments on the manuscript; J Brown, E Carson, S Fong, A McCormick, M Ortiz, J Powell, J Sanders, & D Siegel for technical assistance. R Kiani for creation of scripts for Psychtoolbox used to run the behavioral task. F Molter and A Thomas for creation of the *glambox* toolbox. Funding: SML was supported by the Behavioral and Neural Sciences Graduate Program, the Rutgers University Academic Advancement Fund, and a Dean's Dissertation Fellowship. VBM was supported by the Howard Hughes Medical Institute (WT Newsome), National Institutes of Health Grant K01-DA-036659-01 (VBM), the Busch Biomedical Foundation (VBM), and the Whitehall Foundation (VBM).

## Additional information

### Funding

| Funder | Grant reference number | Author |
|---|---|---|
| Rutgers, The State University of New Jersey | Deans Dissertation Fellowship | Shira M Lupkin |
| Rutgers, The State University of New Jersey | Academic Advancement Fund | Shira M Lupkin |
| Rutgers, The State University of New Jersey | Graduate Assistantship through the Behavioral and Neural Sciences Graduate Program | Shira M Lupkin |
| Whitehall Foundation | | Vincent B McGinty |
| Busch Biomedical Research Foundation | | Vincent B McGinty |
| National Institute on Drug Abuse | K01-DA-036659-01 | Vincent B McGinty |

The funders had no role in study design, data collection and interpretation, or the decision to submit the work for publication.

### Author contributions

Shira M Lupkin, Data curation, Formal analysis, Investigation, Visualization, Methodology, Writing - original draft, Writing – review and editing; Vincent B McGinty, Conceptualization, Data curation, Supervision, Funding acquisition, Investigation, Visualization, Methodology, Project administration, Writing – review and editing

### Author ORCIDs

Shira M Lupkin  http://orcid.org/0000-0002-3792-5571
Vincent B McGinty  http://orcid.org/0000-0003-0883-4301

### Ethics

All procedures were in accordance with the Guide for the Care and Use of Laboratory Animals (2011) and were approved by the Institutional Animal Care and Use Committees of both Stanford University (APLAC Protocol #9720) and Rutgers University-Newark (PROTO999900861). Surgeries to implant orthopedic head restraints were conducted using full surgical anesthesia using aseptic techniques and instruments, and with analgesics and antibiotics given pre-, intra-, and post-operatively as appropriate.

### Decision letter and Author response

Decision letter https://doi.org/10.7554/eLife.78205.sa1
Author response https://doi.org/10.7554/eLife.78205.sa2

---

## Additional files

### Supplementary files

• Supplementary file 1. Mixed effects model specifications. All models included a monkey-specific slope and intercept for each effect. In Wilkinson notation, this is indicated by including (*1+effect(s)* | *monkey*). In line 14, the colon (:) indicates an interaction term.

• Supplementary file 2. Parameter estimates from both GLAM variants. Maximum a posteriori (MAP) and 97.5 highest posterior density (HPD) interval for each parameter obtained from fitting each variant to all data available for each monkey. Note that for "Bias Model", $\gamma$ was a free parameter in the model, while in the "No Bias Model", $\gamma$ was fixed at 1. $\gamma$, gaze bias parameter; $\sigma$, noise parameter; $\tau$, logistic scaling parameter; $\nu$, velocity parameter. **WAIC,** widely applicable information criteria. See Methods for further details on parameter definitions.

• Supplementary file 3. Effect of NDT truncation on relative fit of the GLAM with either an additive or a multiplicative gaze term. Table entries give the WAIC with standard errors in parentheses * indicates models where the standard errors for the two models do not overlap, indicating the fits are

statistically different from each other.

• Supplementary file 4. Comparison of GLAM simulations to held out trials. Estimates and confidence intervals are the result of mixed effects models where each metric was regressed against a binary variable indicating whether the data were from the predicted trials or the observed trials. Confidence intervals including zero indicate no significant difference between predicted vs. observed, i.e. that the predicted data are a good fit to out of sample observations. The choice metric was assessed using a logistic regression. The others were linear. All models included "Monkey" as a random effect. p-values were obtained from an ANOVA on the output of the mixed effects regressions.

• Transparent reporting form

### Data availability

All data and code used for the analyses and figures included in the present manuscript have been uploaded as an Open Science Framework project (and a linked GitHub account). These files can be accessed at: https://osf.io/hkgmn/.

The following dataset was generated:

| Author(s) | Year | Dataset title | Dataset URL | Database and Identifier |
|---|---|---|---|---|
| Lupkin SM, McGinty VB | 2022 | NHP-Gaze-Bias | https://doi.org/10.17605/OSF.IO/HKGMN | Open Science Framework, 10.17605/OSF.IO/HKGMN |

The following previously published dataset was used:

| Author(s) | Year | Dataset title | Dataset URL | Database and Identifier |
|---|---|---|---|---|
| Thomas AW, Molter F, Krajbich I, Heekeren HRH, Mohr PNC | 2019 | Gaze Bias Differences Capture Individual Choice Behavior | https://github.com/glamlab/gaze-bias-differences | GitHub, gaze-bias-differences |

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
