## [Editor Report]

This study analyzed viewing behavior in monkeys during value-based decision-making to determine whether relationships between gaze patterns and choices previously described in humans are also present in monkeys. The study used a clever task design and sophisticated modeling approaches to reveal robust evidence for similarities to extant human data. This is important to the field because it suggests common neural mechanisms linking viewing behavior and decision-making, which can now be further explored across species.

---

## [Decision Letter]

**Decision letter after peer review:**

Thank you for submitting your article "Monkeys exhibit human-like gaze biases in economic decisions" for consideration by *eLife*. Your article has been reviewed by 3 peer reviewers, one of whom is a member of our Board of Reviewing Editors, and the evaluation has been overseen by Michael Frank as the Senior Editor. The reviewers have opted to remain anonymous.

Essential revisions:

Below are three specific points, distilled from the attached reviews, considered essential revisions for the authors to address. The individual reviews appended below can be consulted for further elaboration on these points and additional suggestions to improve the manuscript.

1. A general concern is that the value associated with an item can potentially attract gaze independent of choice, and reviewers felt that these relationships should be considered in more detail (addressed primarily by R1 and R3). On one hand, given a limited set of values, value difference is highly correlated with both the options and the chosen value, and it is suggested that the authors attempt to disentangle these in their results. Similarly, effects on fixation duration should be explored in more detail, including whether they relate to only one of the option values or both, and how they vary when there are 2 versus 3 fixations in a choice. Finally, it was felt that there should be additional discussion of the complex interplay in which a valuable item attracts attention and attention to an item may also increase its subjective value, and the extent to which these possibilities are disentangled or confounded in the current data. (Note that this final concern may be somewhat addressed by the analyses suggested in point 3 below).

2. A second concern is that the approach to "non-decision time" was viewed as arbitrary and perhaps not entirely valid (addressed primarily by R2 and R1). The current approach assumes that final and non-final fixations should be the same length, although other work has shown that final fixations are generally shorter. It also assumes that motor preparation time is equal across all chosen values, although higher values typically motivate faster motor responses. Point 3 below makes a suggestion about how to consider NDT in alternative choice models. Other potential approaches to address this concern within the present model include testing different estimates for NDTs and demonstrating the robustness of the overall results.

3. Finally, reviewers felt that the study would be strengthened by additional exploration of their data in light of studies that assess whether gaze might directly increase drift rate (i.e., gaze is additive to value), rather than (or in addition to) multiplicative effects, where gaze amplifies the attended value. (See Cavanagh et al. 2014 JEP General, Gluth et al. 2018 *eLife*, and Smith and Krajbich 2019 Psych Sci.)

Specifically, Westbrook et al. (Science, 2020) suggested a hybrid two stage model whereby gaze is multiplicative on value early during choice and additive later, with a rationale similar to that discussed here ("post-decisional gaze anchoring" – see also recent work by Callaway et al. 2021 PLoS CB for a related but more continuous model). The central assumption of GLAM is that gaze weights the impact of value on evidence accumulation, but given the issues regarding NDT, considering other models in one way or another is important, and the authors have the trial counts to do so quantitatively. Thus, the authors should address the issue of whether gaze effects in monkeys are multiplicative as assumed by GLAM, or whether they might be additive when applied to the data. There was extensive discussion on this topic in consultation, and indeed there is evidence for both in the human literature. Clearly, publication decisions will not depend on the conclusions of this exploration per se, but a strength of the current data set is that it may be able to speak to this unresolved issue.

One consideration in undertaking such an alternative model analysis is that it could be done with and without truncating NDT (so that NDT would be captured by the second phase of a 2-phase model) and/or by better justifying the choice of NDT. Moreover, it was felt that a model/parameter recovery analysis is a critical component of these explorations. In other words, if Model A "wins", simulated data from model A should be able to reproduce key features of the data that Model B cannot capture, and vice versa and parameter estimates should be recoverable when fitting to those data (see Wilson and Collins 2019 *eLife*).

*Reviewer #1 (Recommendations for the authors):*

Critique 1: The authors should consider the possibility that unique option values can explain putative effects of value difference. One approach might be to subselect trials and analyze data sets where chosen value varies but value difference is constant and vice versa.

Critique 2: The concern about whether effect first fixated values carry over into behavior during the second fixation could be addressed similarly to critique 1 or with model comparison approaches.

Critique 3: The authors should consider whether a more flexible definition of NDTs could better fit behavior. This concern might also be addressed in approaches to critique 4.

Critique 4: The concern in this critique is that value and choice can have influence on gaze behavior (and drift rates) that is not accounted for by the present models. Addressing this would involve more extensive re-analysis than the previous points, however given that the contribution of this paper is showing that monkey behavior recapitulates what has been reported in humans, I think it is warranted to explore more complex interactions between gaze and choice that have been found in humans subjects. To address this concern, the authors should consider comparing the present results to models that allow the γ parameter to vary with specific task parameters (for instance as in Westbrook, 2020).

*Reviewer #2 (Recommendations for the authors):*

It seems a little silly to expend all this effort generating bootstrapped distributions of final and non-final fixation differences, only to then arbitrarily choose the 95th percentile of that distribution as the terminal non decision time. Why not instead either simply get the best estimate you can from the literature, and/or demonstrate the robustness of the results to different non-decision time corrections (e.g. 0, 200, 400 ms)?

Regarding the manipulation where onsets were staggered: the analysis here seemed convoluted. Why not directly test whether subjects were more likely to choose the initially presented option? Figure 3 helps to address this, but leaves some ambiguity for Monkey K, who doesn't always initially fixate on the first presented alternative.

It may be worth noting that the effects in Figure 5 appear to be smaller than in most human data. This might be an argument for 'not' cutting the last 200ms of each trial?

On page 28 the authors discuss Krajbich et al. 2021 and argue that their findings are consistent with LIP, frontal eye fields, and the superior colliculus accumulating evidence in studies where eyes are used to report decisions. However, that study was like this one; subjects were free to look around but they chose using the keyboard.

I thought the authors missed an opportunity to discuss some of the human neuroimaging work on value-based SSM, given the focus of their Discussion section. There are a number of well-known articles that are relevant, for example:

Hare et al. 2011 (PNAS); Gluth et al. 2012 (J Neuro); Rodriguez et al. 2015 (European J Neurosci); Pisauro et al. 2017 (Nature Comm).

Why did the authors use different delays for the two monkeys?

How long were the delays between releasing the center lever and pressing left or right?

Was there some criterion for passing the training stage?

p.33-34 – these numbers should go in the Results section, not the Methods.

I think it might make sense to include session-level random effects in the models, given that each session used a different set of stimuli. I doubt this will change much, but it might further help clean up some variance.

p.38 – This argument about how early gaze might be used to explore and evaluate targets while late gaze is used to focus on the to-be-chosen target comes nearly straight out of Westbrook et al. 2020 (Science).

*Reviewer #3 (Recommendations for the authors):*

I would have liked to see a more extensive discussion of what might underly the observed effects. The sequential sampling models are merely descriptive. One would like to know how fixation influences the neural signals that compute the value of the option. When a human/monkey is looking at one option, information about a previously viewed option must be maintained in working memory and the neural signals might be expected to be noisier, leading to a bias towards the most recently viewed option. Similarly, the first viewed option might have a stronger signal than the second viewed item by virtue of having less interference. This is obviously speculative but it would be nice to see what evidence there is for something like this – indeed the McGinty et al. 2016 paper seems to provide evidence of this kind. One could argue that it shouldn't be too surprising that computations performed at the fixation point are higher S/N than ones off fixation. (In the context of the sequential sampling model this would correspond to adding some noise to the item not currently fixated.)

Related to this, the discussion of how the paradigm might be used for further exploration is a bit vague. The example given of MT and LIP is not particularly well chosen, given the recent evidence from the Huk lab (Katz et al., Nature 2016) against LIP as the site of the perceptual decision.

Another general concern is the absence of any discussion of all the work showing that value associated with an item serves to attract gaze and indeed is a central aspect of the mechanisms of gaze control and learning where to attend. One closely related example is the work of Hikosaka (eg Kim et al., Cell, 2015) showing that cells in the caudate tail coded the value of previously rewarded fractal patterns and attracted gaze. This makes the question of causality a bit difficult to disentangle. It's such a central issue that it needs to be explicitly discussed. Having the stimuli not resolvable in the peripheral retina helps with this issue.

I would also like to see more of the actual fixation time data. If there are only two fixations, having a bias towards the first one viewed is in conflict with a bias towards the last one viewed. Thus it would be nice to see the fixation durations broken down into cases where there are 2 fixations and cases where there are 3 fixations. Just as the calculation that the initial-view bias is equal to about half a drop of juice, it would also be useful to know how fixation duration translates to choice bias. If I am understanding Figure 4 correctly it looks like an extra 200 msec viewing time translates to about a 10% increase in choice probability. More concrete description of the fixations would be helpful. For example, how do the β values on p8-9 translate to fixation durations?

---

## [Author Response]

Essential revisions:Reviewer #1 (Recommendations for the authors):Critique 1: The authors should consider the possibility that unique option values can explain putative effects of value difference. One approach might be to subselect trials and analyze data sets where chosen value varies but value difference is constant and vice versa.

To address this question, we have adapted the methods of Balewski and colleagues (*Neuron*, 2022) to isolate the unique contributions of chosen value and trial difficulty to reaction time and the number of fixations in a given trial (the two behaviors modulated by difficulty in the original paper). This new analysis reveals a double dissociation in which reaction time decreases as a function of chosen value but not difficulty, while the number of fixations in a trial shows the opposite pattern. Our interpretation is that reaction time largely reflects reward anticipation, whereas the number of fixations largely reflects the amount of information required to render a decision (i.e., choice difficulty). See lines 144-167 and Figure 2.

Critique 2: The concern about whether effect first fixated values carry over into behavior during the second fixation could be addressed similarly to critique 1 or with model comparison approaches.

This is a valid interpretation of the results. To test this directly, we now include an analysis of middle fixation duration as a function of the not-currently viewed target. Note that the vast majority of middle fixations are the second fixation in the trial, and therefore the value of the unattended target is typically the one that was viewed first. The analysis showed a negative correlation between middle fixation duration and the value of the unattended target which is consistent with the first fixated value carrying over to the second fixation. See lines 243-246.

Critique 3: The authors should consider whether a more flexible definition of NDTs could better fit behavior. This concern might also be addressed in approaches to critique 4.

In all sequential sampling model formulations we are aware of, nondecision time is considered to be fixed across trial types. Examples can be found for perceptual decisions (e.g., Resulaj et al., 2009) and in the “bifurcation point” approach used in the recent value-based decision study by Westbrook et al. (2020).

To further investigate this issue, we asked whether other post-decision processes were sensitive to chosen value in our paradigm. To do so, we measured the interval between the center lever lift and the left or right lever press, corresponding to the time taken to perform the reach movement in each trial (reach latency). We then fit a mixed effects model explaining reach latency as a function of chosen value. While the results showed significantly faster reach latencies with higher chosen values, the effect size was very small, showing on average a ~3ms decrease per drop of juice. In other words, between the highest and lowest levels of chosen value (5 vs. 1), there is only a difference of approximately 12ms. In contrast, the main RT measure used in the study (the interval between target onset and center lever lift) is an order of magnitude more sensitive to chosen value, decreasing ~40ms per drop of juice. These results are shown in Author response image 1.

**Author response image 1. sa2fig1:** 

This suggests that post-decision processes (NDT in standard models and the additive stage in the Westbrook paper) vary only minimally as a function of chosen value. We are happy to include this analysis as a supplemental figure upon request.

Critique 4: The concern in this critique is that value and choice can have influence on gaze behavior (and drift rates) that is not accounted for by the present models. Addressing this would involve more extensive re-analysis than the previous points, however given that the contribution of this paper is showing that monkey behavior recapitulates what has been reported in humans, I think it is warranted to explore more complex interactions between gaze and choice that have been found in humans subjects. To address this concern, the authors should consider comparing the present results to models that allow the γ parameter to vary with specific task parameters (for instance as in Westbrook, 2020).

The two-stage model of gaze effects put forth by Westbrook et al. (2020) is consistent with other observations of gaze behavior and choice (i.e., Thomas et al., 2019, Smith et al., 2018, Manohar and Husain, 2013). In this model, gaze effects early in the trial are best described by a multiplicative relationship between gaze and value, whereas gaze effects later in the trial are best described with an additive model term. To test the two-stage hypothesis, Westbrook and colleagues determined a ‘bifurcation point’ for each subject that represented the time at which gaze effects transitioned from multiplicative to additive. In our data, trial durations were typically very short (<1s), making it difficult to divide trials and fit separate models to them. We therefore took at different approach: We reasoned that if gaze effects transition from multiplicative to additive at the end of the trial, then the transition point could be estimated by removing data from the end of each trial and assessing the relative fit of a multiplicative vs. additive model. If the early gaze effects are predominantly multiplicative and late gaze effects are additive, the relative goodness of fit for an additive model should decrease as more data are removed from the end of the trial. To test this idea, we compared the relative model fit of an additive vs. multiplicative models in the raw data, and for data in which successively larger epochs were removed from the end of the trial (50, 100, 150, 200, 300, and 400ms). The relative fit was assessed by computing the relative probability that each model accurately reflects the data. In addition, to identify significant differences in goodness of fit, we compared the WAIC values and their standard errors for each model (Supplementary file 3). As shown in Figure 4, the relative fit probability for both models is nonzero in the raw data 0 truncation, indicating that neither model provides a definitive best fit, potentially reflecting a mixture of the two processes. However, the relative fit of the additive model decreases sharply as data is removed, reaching zero at 100ms truncation. 100ms is also the point at which multiplicative models provide a significantly better fit, indicated by non-overlapping standard error intervals for the two models (Supplementary file 3). Together, this suggested that the transition between early- and late-stage gaze effects likely occurs approximately 100ms before the RT.

To minimize the influence of post-decision gaze effects, the main results use data truncated by 100ms. However, because 100ms is only an estimate, we repeated the main analyses over truncation values between 0 and 400ms, reported in Figure 6 —figure supplement 1 and Figure 7 —figure supplement 1. These show significant gaze duration biases and final gaze biases in data truncated by up to 200ms.

Reviewer #2 (Recommendations for the authors):It seems a little silly to expend all this effort generating bootstrapped distributions of final and non-final fixation differences, only to then arbitrarily choose the 95th percentile of that distribution as the terminal non decision time. Why not instead either simply get the best estimate you can from the literature, and/or demonstrate the robustness of the results to different non-decision time corrections (e.g. 0, 200, 400 ms)?

Thanks for the opportunity to clarify these points. There are three related issues:

First, with regards to fixation durations, in the updated Figure 3 we now show durations as a function of both the absolute order in the trial (first, second, third, fourth, etc.) and the relative order (final/nonfinal). We find that durations decrease as a function of absolute order in the trial, an effect also seen in humans (see Manohar and Husain, 2013). At the same time, while holding absolute order constant, final fixations are longer than non-final fixations. To explain the discrepancy with human final fixation durations, we note that monkeys make many fewer fixations per trial (~2.5) than humans do (~3.7, computed from publicly available data from Krajbich et al., 2010.) This means that compared to humans, monkeys’ final fixations occur earlier in the trial (e.g., second or third), and are therefore comparatively longer in duration. Note that studies with humans have not independently measured fixation durations by absolute and relative order, and therefore would not have detected the potential interaction between the two effects.

Second, the comment suggests that the final 200ms before lever lift is not spent planning the left/right movement, given that the monkeys have time *after* the lever lift in which to execute the movement (400 or 500ms, depending on the monkey). The presumption appears to be that 400/500ms should be sufficient to plan a left/right reach. However, we think that these two suggestions are unlikely, and that our original interpretation is the most plausible. First, the 400/500ms deadline between lift and left/right press was set to encourage the monkeys to complete the reach as fast as possible, to minimize deliberations or changes of mind after lifting the lever. More specifically, these deadlines were designed so that on ~0.5% of trials, the monkeys actually fail to complete the reach within the deadline and fail to obtain a reward. This manipulation was effective at motivating fast reaches, as the average reach latency (time between lift and press) was 165 SEM 20ms for Monkey K, and 290 SEM 100ms for Monkey C.

Therefore, given the time pressure imposed by the task, it is very unlikely that significant reach planning occurs after the lever lift. In addition to these empirical considerations, the idea that the final moments before the RT are used for motor planning is a standard assumption in many theoretical models of choice (including sequential sampling models, see Ratcliff and McKoon 2008, for review), and is also well-supported by studies of motor control and motor system neurophysiology.

Based on these, we think the assumption of some form of terminal NDT is warranted.

Third, we have changed our method for estimating the NDT interval. In brief we sweep through a range of NDT truncation values (0-400ms) and identify the smallest interval (100ms) that minimizes the contribution of “additive” gaze effects, which are thought to reflect late-stage, post-decision gaze processes. See the response to Point 4 for Reviewer 1, Figure 4 and lines 267-325 in the main text. In addition, we report all of the major study results over a range of truncation values between 0 and 400ms.

Regarding the manipulation where onsets were staggered: the analysis here seemed convoluted. Why not directly test whether subjects were more likely to choose the initially presented option? Figure 3 helps to address this, but leaves some ambiguity for Monkey K, who doesn't always initially fixate on the first presented alternative.

First, to eliminate ambiguity in the interpretation of the initial fixation bias for the manipulation sessions, we have removed the 99 trials (1.82%) in Monkey K’s data where he did not look at the suggested first target. We have changed the intext statistics and the corresponding figure, accordingly (now Figure 6). Removing these trials did not change any of the conclusions.

Second, we would like to take the opportunity to explain the rationale for the analysis of the gaze manipulation sessions. In the primary dataset (i.e., sessions without manipulation), we found that the monkeys were more likely to choose the first target that they looked at – evident by simply dividing the trials as the comment suggests.

The purpose of the manipulation sessions was to test the *causal* nature of the initial fixation bias by staggering the onset of the targets on a fraction of trials. Additionally, we also wanted to assess whether the size of the bias on the manipulation trials differed from the size of the bias in the non-manipulation trials collected in the same sessions.

To that end, for the data from the manipulation sessions we fit a single linear model that explains the fraction of left choices as a function of: (1) the value difference between the left and right targets; (2) the direction of initial fixation (left/right); (3) a Boolean variable indicating whether or not a given trial contained gaze manipulation; and (4) the interaction between the trial type and the location of the initial fixation. In Wilkinson notation, this can be written as:

P(choose-left) ~ value-difference + first-is-left + trial-type + (first-is-left:trial-type).

The first-is-left regressor indicates whether there was an overall effect of initial fixation location on the fraction of left choices (the main effect of interest). The trialtype regressor indicates whether the overall fraction of left choices differed between trials with or without gaze manipulation. And the interaction term indicates whether the effect of the initial fixation differed according to the trial-type – i.e., depended on whether the location of the first fixation was dictated to the monkey or if he was able to choose where to deploy his first fixation.

As detailed in lines 349-361, this analysis identified a significant main effect of firstis-left, and no significant effects for trial-type or the interaction, indicating that the bias due to initial gaze location was no different in trials with vs. without gaze manipulation. This analysis provides a parsimonious explanation of the effects of initial gaze during the manipulation sessions.

Note that, except for the trial-type regressor and the interaction term, this is the same model used to describe the initial fixation bias in the sessions without the any gaze manipulation, as well as the one used to assess this bias in humans (Krajbich et al., 2010).

It may be worth noting that the effects in Figure 5 appear to be smaller than in most human data. This might be an argument for 'not' cutting the last 200ms of each trial?

The reviewer’s intuition is correct that there is a relationship between the NDT truncation and the final fixation bias. As shown in Figure 7 —figure supplement 1, we find that the final fixation bias decreases as more time is truncated from the end of the trial.

The responses to Reviewer 1 (Points 3 and 4) and Reviewer 2 (Point 1), articulate the motivation for estimating terminal NDT and removing time from the end of each trial. In a nutshell, terminal non-decision time is standard assumption in many sequential sampling models, and is supported by evidence from motor control and motor system neurophysiology. In addition, our data appear consistent with a two stage process in which gaze becomes drawn to the to-be-chosen target within in the final 100ms of a trial Figure 4 and lines 267-325 Because our goal is to observe the early-stage gaze effects, we truncate the data to minimize the influence of late-stage gaze effects.

The revised manuscript uses an NDT estimate of 100ms for main figures (Figures 59) and analyses, and also reports main results for data truncated over a range from 0 to 400ms (Figure 6 —figure supplement 1 and Figure 7 —figure supplement 1.). The magnitude of gaze bias effects using 100ms truncation is similar to those reported in human studies.

On page 28 the authors discuss Krajbich et al. 2021 and argue that their findings are consistent with LIP, frontal eye fields, and the superior colliculus accumulating evidence in studies where eyes are used to report decisions. However, that study was like this one; subjects were free to look around but they chose using the keyboard.I thought the authors missed an opportunity to discuss some of the human neuroimaging work on value-based SSM, given the focus of their Discussion section. There are a number of well-known articles that are relevant, for example:Hare et al. 2011 (PNAS); Gluth et al. 2012 (J Neuro); Rodriguez et al. 2015 (European J Neurosci); Pisauro et al. 2017 (Nature Comm).

Thank you for the suggestions. We did not intend to suggest that participants in the Krajbich 2021 study indicated choice with their eyes. The relevant part of the discussion has been updated, as follows:

“For the encoding of accumulated evidence, findings from NHP neurophysiology point to regions involved in the preparation of movement. In motion discrimination tasks that use eye movements to report decisions, accumulated evidence signals can be observed in oculomotor control regions such as area LIP, frontal eye fields, and the superior colliculus. Because the decision in our task is reported with a reach movement, potential sites for accumulator-like activity include motor cortical areas such as the arm representations within dorsal pre-motor cortex (Chandrasekaran et al., 2017). In addition, human imaging studies have identified several other candidate regions with accumulator-like activity, including in the intraparietal sulcus, insula, caudate, and lateral prefrontal cortex (Gluth et al., 2012; Hare et al., 2011; Pisauro et al., 2017; Rodriguez et al., 2015). Interestingly, several studies find accumulator-like signals in cortical areas in the dorsal and medial frontal cortex, but these are not consistently localized. For example, whereas Pisauro et al. (2017) find accumulator like activity in the supplementary motor area, Rodriguez et al. (2015) identify a more anterior region in the medial frontal cortex. Additional studies in both humans and NHPs will be necessary to understand the specializations of these regions with respect to encoding decision evidence.”

Why did the authors use different delays for the two monkeys?

This comment seems to refer to the delay between the left/right lever press and the juice delivery, which was either 1 or 1.5s depending on the monkey. In our experience, introducing a delay between response and reward slows the overall reward rate that the animal experiences, which encourages slower reaction times and more accurate choices. The exact duration of this delay was tuned for each monkey during training – hence the different delays used for monkeys K and C. Because this delay occurs after the response lever is pressed, it is inconsequential for the study results.

How long were the delays between releasing the center lever and pressing left or right?

After lifting the center lever, Monkey C had up to 500ms and Monkey K had up to 400ms to press the left or right lever. These deadlines were determined individually for each monkey during their training and were set so that on a very small fraction of trials (~0.5%) the monkey failed to press the lever in time. This ensures that the monkey completes the responses quickly and minimizes opportunity for deliberation or changes of mind. On average, Monkey C’s lift-to-press latency was 290ms SEM 100ms and Monkey’s K’s average was 165ms SEM 20ms.

Was there some criterion for passing the training stage?

The criteria were qualitative. When the monkeys were first learning the fundamentals of the task, they were trained until the monkey could reliably learn stimulus-reward associations for a new stimulus set within several sessions, evident in their choice and reaction time curves (as in Figure 1D/E). At that point, we continued to introduce a new stimulus set every 2-5 sessions. When a new set was introduced, we considered the set to be “learned” when across the most recent ~200 trials the choices for the easiest stimuli (+/-4) were virtually 100% correct, choices for the next easiest (+/- 3) were ~80% correct or above, and the RT curves had a clearly visible negative slope (as in Figure1E). In fully trained animals, this threshold was usually met after 1-2 sessions. Data collection for each new set only began after this threshold was met, and each set was used for 1-3 data collection sessions.

p.33-34 – these numbers should go in the Results section, not the Methods.

These statistics are now also found on line 121.

I think it might make sense to include session-level random effects in the models, given that each session used a different set of stimuli. I doubt this will change much, but it might further help clean up some variance.

First, we’d like to clarify that each stimulus set was used for 1-3 consecutive sessions (preceded by 1-2 training sessions not used for analysis).

We re-ran the key analyses using either session, or the session-by-monkey interaction as a random effect. For both of these alternative models, the results were very similar to the model with Monkey as a random effect, and did not suggest any changes in the interpretation of the findings (not shown).

The Akaike Information Criterion (AIC) for each of the three models is presented in Author response table 1, with the ‘winning’ model (lowest AIC) for each analysis marked with’*’. On the whole, the data were better fit by the monkey-only random effect model; this is likely due to the low intersession variability that results from extensive training. Consequently, the additional terms required by the session or session-by-monkey random effect model variants (54 levels of random effects per regressor vs. 2 levels) is disproportionately larger than the amount of additional variance explained—leading to higher AIC values. For this reason, the monkey-specific random effects models are the most appropriate, and we report these results in the manuscript. The results of alternative models can be provided as supplements if requested.

**Author response table 1. sa2table1:** 

Analysis	Session RE	Monkey:Session RE	Monkey RE
Choice	1.68E+05	1.67E+05	1.64E+05*
RT	-4.06E+04	-4.06E+04	-3.68E+04*
Num. Fix	3.32E+04*	3.32E+04*	3.33E+04
Initial Fixation Bias	1.70E+05	1.70E+05	1.66E+05*
Final Fixation Bias	1.60E+05	1.60E+05	1.57E+05*
Cumulative Gaze Time-Bias	1.25E+05	1.25E+05	1.24E+05*
Cumulative Gaze-Time Bias :Corrected	1.87E+04*	1.87E+04*	1.90E+04

Reviewer #3 (Recommendations for the authors):I would have liked to see a more extensive discussion of what might underly the observed effects. The sequential sampling models are merely descriptive. One would like to know how fixation influences the neural signals that compute the value of the option. When a human/monkey is looking at one option, information about a previously viewed option must be maintained in working memory and the neural signals might be expected to be noisier, leading to a bias towards the most recently viewed option. Similarly, the first viewed option might have a stronger signal than the second viewed item by virtue of having less interference. This is obviously speculative but it would be nice to see what evidence there is for something like this – indeed the McGinty et al. 2016 paper seems to provide evidence of this kind. One could argue that it shouldn't be too surprising that computations performed at the fixation point are higher S/N than ones off fixation. (In the context of the sequential sampling model this would correspond to adding some noise to the item not currently fixated.)

Thank you for the suggestion. The following has been added to the discussion.:

“In NHPs, the orbitofrontal portion of the VMF (the OFC) has been demonstrated to contain value-signals that are dependent on whether gaze is directed towards a reward-associated visual target (Hunt et al., 2018; McGinty et al., 2016; McGinty and Lupkin, 2021) a mechanism fully consistent with the effects of gaze posited by aSSM frameworks. However, the precise neural mechanisms explaining how gaze biases choice outcomes are still unknown. One hypothesis is that the value-coding regions inherit gaze-modulated information from visual cortical regions sensitive to shifts in visual attention. For instance, neurons in the anterior inferotemporal cortex, which projects directly to the OFC (Carmichael and Price, 1995; Kravitz et al., 2013), selectively encode features of attended objects to the exclusion of others in the same receptive field (DiCarlo and Maunsell, 2000; Moore et al., 2003; Moran and Desimone, 1985; Richmond et al., 1983; Sheinberg and Logothetis, 2001). Therefore, the similar fixation-driven effects on OFC value-coding shown by McGinty and colleagues (McGinty et al., 2016; McGinty and Lupkin, 2021) may be inherited from this structure. Another candidate region for the top-down influences on gaze dependent value-coding, is the frontal eye fields (FEF). In a recent study from Krajbich et al. (2021), focal disruption of FEF leads to a reduction in the magnitude of gaze-biases. Prior work in both humans and NHPs has shown that the FEF can influence perceptual sensitivity through the top-down modulation of early visual areas (Moore and Armstrong, 2003; Moore and Fallah, 2004; Ruff et al., 2006; Silvanto et al., 2006; Taylor et al., 2007), suggesting that it may exert similar influence on value encoding regions.”

Related to this, the discussion of how the paradigm might be used for further exploration is a bit vague. The example given of MT and LIP is not particularly well chosen, given the recent evidence from the Huk lab (Katz et al., Nature 2016) against LIP as the site of the perceptual decision.

Thank you for the suggestion. The discussion has been updated as follows:

“The impetus for developing this animal model is to explore the neural underpinnings of the relationship between gaze and value-based choice. Given the fact that a sequential sampling model provides objectively good fits to the observed behavior, one potential approach for understanding neural mechanisms is to identify brain regions that correspond to different functions in a sequential sampling framework. Such an approach has been used in perceptual dot motion discrimination, where extrastriate cortical area MT is thought to provide the “input” signal (encoding of visual motion) and neurons in parietal area LIP are thought to represent the accumulated evidence over time (see Gold and Shadlen, 2007, for review). One note of caution, however, is that brain activity that outwardly appears to implement accumulator-like functions may not be causally involved in the choice (e.g. Katz et al., 2016). Nonetheless, it is still useful to consider the neural origins of two core computations suggested by sequential sampling models: the representation of the target values (corresponding to the input signals), and neural signals that predict of the actions performed to obtain the reward (reflecting the decision output).”

Another general concern is the absence of any discussion of all the work showing that value associated with an item serves to attract gaze and indeed is a central aspect of the mechanisms of gaze control and learning where to attend. One closely related example is the work of Hikosaka (eg Kim et al., Cell, 2015) showing that cells in the caudate tail coded the value of previously rewarded fractal patterns and attracted gaze. This makes the question of causality a bit difficult to disentangle. It's such a central issue that it needs to be explicitly discussed. Having the stimuli not resolvable in the peripheral retina helps with this issue.

The discussion has been updated as follows:

“Our study confirms this result, and provides additional insights, due to novel elements of the study design. First, by manipulating initial fixation direction, we show that initial fixation has a causal effect on the decision process. Second, the results of the aSSM model-fitting provide a mechanistic explanation for our results and permits a direct comparison to model fitting results in human studies.

Finally, unlike many prior studies, the decision task requires the monkeys to sample the targets by fixating them directly, due to the use of a gaze contingent mask that completely obscures the stimuli at the beginning of the trial, and the use of visual crowders that closely surround each target. This subtle design element is crucial, because it permits accurate measurement of the time spent attending to each target, and of the relationship between relative viewing time and choice. In a forthcoming study using this same task, we show that OFC neurons do not begin to reflect the value of the second-fixated stimulus until well after it has been viewed by the monkey, indicating that value information is only available to the monkey once a target is fixated (McGinty and Lupkin, 2021). In contrast, in tasks where value associated targets are easily perceived in peripheral vision, even targets that are not fixated influence prefrontal value signals and decision behavior (e.g., Cavanagh et al., 2019; Xie et al., n.d.). High-value peripheral targets are also implicated in valuebased attentional capture, in which overt or covert attention becomes drawn to objects that are reliably associated with reward (Anderson et al., 2011; Kim et al., 2015). Thus, the task used in the present study minimizes value-based capture effects.”

I would also like to see more of the actual fixation time data. If there are only two fixations, having a bias towards the first one viewed is in conflict with a bias towards the last one viewed. Thus it would be nice to see the fixation durations broken down into cases where there are 2 fixations and cases where there are 3 fixations. Just as the calculation that the initial-view bias is equal to about half a drop of juice, it would also be useful to know how fixation duration translates to choice bias. If I am understanding Figure 4 correctly it looks like an extra 200 msec viewing time translates to about a 10% increase in choice probability. More concrete description of the fixations would be helpful. For example, how do the β values on p8-9 translate to fixation durations?

A detailed description of fixation durations as a function of absolute trial order and relative trial order (final vs. non-final fixations) is shown in Figure 3 and described in lines 196-233. Intuitive descriptions of the choice bias effects, and fixation duration effects have been added on lines 386-388 and 238-243, respectively.